# Delegation and Verification under AI

Lingxiao Huang [1]   Wenyang Xiao [1]   Nisheeth K. Vishnoi [2]

## Abstract

As AI systems enter institutional workflows, workers must decide whether to delegate task execution to AI and how much effort to invest in verifying AI outputs, while institutions evaluate workers using outcome-based standards that may misalign with workers' private costs. We model delegation and verification as the solution to a rational worker's optimization problem, and define worker quality by evaluating an institution-centered utility (distinct from the worker's objective) at the resulting optimal action. We formally characterize optimal worker workflows and show that AI induces *phase transitions*, where arbitrarily small differences in verification ability lead to sharply different behaviors. As a result, AI can amplify workers with strong verification reliability while degrading institutional worker quality for others who rationally over-delegate and reduce oversight, even when baseline task success improves and no behavioral biases are present. These results identify a structural mechanism by which AI reshapes institutional worker quality and amplifies quality disparities between workers with different verification reliability.

## 1. Introduction

Across high-stakes domains such as medicine, finance, and law, task completion traditionally followed a simple structure: a human worker executed the task and bore responsibility for correctness. As artificial intelligence (AI) systems enter institutional workflows, execution can be delegated while responsibility remains human. Institutions increasingly expect workers to use AI for efficiency, standardization, or cost reduction, transforming task completion into a process of delegation and oversight (Mayer et al., 2025; Zi-

rar et al., 2023). Work is therefore organized as a *delegation pipeline*: a worker may perform the task manually, delegate execution to AI, or delegate while verifying the AI's output.

In experimental and real-world settings, workers tend to rely more heavily on AI advice on harder and less predictable tasks, even when AI accuracy is lower in those regimes (Bogert et al., 2021; Passi & Vorvoreanu, 2022; Zhang & Deniz, 2024). At the same time, verification and oversight often weaken under cognitive load, complexity, or time pressure, leading to reduced scrutiny of AI outputs precisely where failures are most likely (Gaube et al., 2021; Bergman, 2025). Together, these findings point to a structural tension between delegation and verification in AI-assisted work. Notably, recent empirical evidence suggests that AI-assisted work can unevenly affect worker outcomes, with some workers benefiting from AI access while others experience degraded performance despite similar exposure to AI tools (Brynjolfsson et al., 2025).

These observations raise a basic question: *How does AI reshape the value that institutions derive from human workers?* We refer to this value as *worker quality*: the institutional utility generated by a worker's output, accounting not only for task success but also for verification costs and accountability borne by the institution. While workers choose among manual work, verified delegation, and pure delegation to optimize their own effort–utility trade-offs, institutional evaluation is typically outcome-based and agnostic to internal workflows. This misalignment implies that individually rational delegation strategies need not maximize institutional utility, motivating the framework we develop in this paper.

**Related work.** Our setting relates to prior work on human–AI delegation, reliance, and monitoring under costly verification, which differ in how delegation and verification decisions are modeled and in whether worker behavior and institutional outcomes are treated endogenously. A growing body of work studies how tasks should be allocated between humans and AI systems, including selective prediction, learning-to-defer, and complementary expertise frameworks (Madras et al., 2018; Cortes et al., 2016; Mozannar & Sontag, 2020; Hemmer et al., 2023). These approaches design routing or deferral policies to optimize system-level accuracy or utility, typically assuming fixed human accuracy

[1]State Key Laboratory of Novel Software Technology, Nanjing University, Nanjing 210023, China [2]Yale University. Correspondence to: Lingxiao Huang <huanglingxiao@nju.edu.cn>, Nisheeth K. Vishnoi <nisheeth.vishnoi@gmail.com>.

*Proceedings of the 43$^{rd}$ International Conference on Machine Learning*, Seoul, South Korea. PMLR 306, 2026. Copyright 2026 by the author(s).

and exogenous human effort, and therefore do not model how worker-chosen delegation interacts with verification effort or translates into institutional worker quality.

Empirical work documents patterns of over-reliance and oversight decay in AI-assisted settings as functions of task difficulty, uncertainty, and cognitive load (Parasuraman et al., 2000; Zhang & Deniz, 2024; Lyell & Coiera, 2017). While this evidence shows that verification weakens in practice, it typically explains these effects in psychological or organizational terms rather than modeling the underlying decision problem linking effort costs, task difficulty, and outcomes.

Economic models of monitoring and verification study how costly inspection or auditing should be deployed to deter misreporting or errors (Townsend, 1979; Gale & Hellwig, 1985; Mookherjee & Png, 1989; Georgiadis & Szentes, 2020). In these models, verification is controlled by an institutional principal, rather than chosen by workers on a per-task basis, and human and AI performance are not jointly modeled as functions of task hardness.

Recent work also develops mathematical frameworks for human–AI integration, characterizing how human and AI capabilities combine to affect job success and productivity (Celis et al., 2025), as well as benchmark- and value-of-information–based approaches to joint decision-making (Kleinberg et al., 2018; Guo et al., 2024; 2025). These frameworks characterize optimal decisions given available signals, but do not model how delegation and verification choices are shaped by effort costs or how they induce institutional outcomes. Taken together, existing approaches address task allocation, reliance behavior, auditing, or evaluation in isolation, but do not provide a unified account of how worker-chosen delegation and verification under AI shape institutional worker quality.

**Our contributions.** We introduce a mathematical framework that characterizes how AI reshapes institutional worker quality through delegation and verification. Our main contributions are as follows:

- We model worker–AI interaction via a delegation–verification action pair $(d, s) \in [0, 1]^2$, where $d$ is the probability of delegating execution to AI and $s$ is verification effort. Worker ability is parameterized by verification reliability $\alpha \in \mathbb{R}_{\geq 0}$ and execution efficiency $\beta \in \mathbb{R}_{\geq 0}$. Worker behavior is given by the solution to a utility maximization problem (Eq. (2)), while *worker quality* is defined by evaluating an institution-centered utility at the resulting optimal action (Eq. (4)). This framework makes explicit the structural misalignment between worker incentives and institutional objectives.
- We characterize the worker's optimal workflow $(d^\star, s^\star)$ as a function of ability parameters $(\alpha, \beta)$ (Theorem 3.1).

We show how the model partitions the ability space into threshold behavior regions, with sharp transitions between manual work, pure delegation, and verified delegation. These transitions arise from structural properties of the delegation pipeline, persist under general concave detection and convex cost functions, and explain why accountability failures can emerge abruptly rather than gradually in AI-assisted workflows.

- We characterize when AI assistance improves or degrades institutional worker quality (Theorem 3.5) and identify which workers are upgraded or downgraded relative to the pre-AI baseline (Theorem 3.6). We identify a *compliance gain* regime, in which workers with high verification reliability become institutionally qualified due to AI access, and a *compliance loss* regime, in which workers with low verification reliability rationally over-delegate and experience quality degradation. These effects arise endogenously from rational optimization, without invoking behavioral bias or miscalibration. More broadly, our results suggest a *verification amplification* phenomenon: AI access can disproportionately benefit workers with strong evaluative capabilities, while disadvantaging others through rational over-delegation.
- In a tractable instantiation with inverse-linear verification reliability and linear execution cost, we obtain closed-form expressions for optimal actions and resulting worker quality (Figure 1). This highlights the central role of verification reliability and reveals countervailing effects: higher verification ability or higher AI quality can induce over-delegation and reduce quality for some workers.
- We formulate a constrained optimization problem for worker up-skilling that trades off improvements in verification reliability $\alpha$ and execution efficiency $\beta$ under a qualification constraint (Eq. (17), Section B.1). We also analyze institutional interventions, including improvements in AI capability and changes to worker incentives, and show that both can have non-monotonic effects on institutional utility (Section 4 and Section B.2).

We further examine the robustness of the model under heterogeneous task difficulty, miscalibrated beliefs about AI capability, and partial re-execution, showing that the core qualitative phenomena persist (Section 4 and Section C). We then provide an illustrative application using real-world data to demonstrate the framework's empirical relevance (Section 5).

Overall, our framework shows that AI functions not only as a productivity enhancer but as a structural filter that induces discontinuous changes in institutional worker quality. By isolating rational over-delegation as the core mechanism, we show that verification ability, rather than execution efficiency, becomes the primary determinant of institutional viability in AI-assisted workflows.

## 2. Model

We study a static, per-task model of AI-assisted work in which a rational worker interacts with a fixed AI system, choosing delegation and verification effort, while the institution evaluates the worker based on task outcomes.

**Action space and workflow.** A worker $W$ chooses an action $(d, s) \in [0, 1]^2$, where $d$ denotes the probability of delegating task execution to AI and $s$ denotes the verification effort exerted upon delegation. With probability $1 - d$, the worker executes the task independently. When the task is delegated, the worker verifies the AI's output with effort $s$. If an error is detected, the worker redoes the task independently of the AI's output. The pre-AI workflow corresponds to $(d, s) = (0, 0)$, in which the worker always executes the task directly.

**Worker utility.** The worker chooses an action to trade off task success against completion cost, which we model via a utility function depending on the probability of task success and the cost incurred. Let $p : [0, 1]^2 \to [0, 1]$ denote the *task success function* and $\text{cost} : [0, 1]^2 \to \mathbb{R}_{\geq 0}$ the *cost function*. For an action $(d, s)$, $p(d, s)$ is the probability of success and $\text{cost}(d, s)$ the incurred cost. Both functions are increasing in verification effort $s$ and depend on worker and AI abilities, as specified in Equations (5) and (6). Let $b_W \geq 0$ denote the worker's benefit from success and $\ell_W \geq 0$ the loss from failure. The worker utility is

$$U_W(d, s) := b_W \, p(d, s) - \ell_W \big(1 - p(d, s)\big) - \text{cost}(d, s). \tag{1}$$

The worker selects an optimal action

$$(d^\star, s^\star) := \arg \max_{(d,s) \in [0,1]^2} U_W(d, s). \tag{2}$$

Since AI enlarges the action space, the worker's utility weakly improves relative to the no-AI baseline: $U_W(d^\star, s^\star) \geq U_W(0, 0)$.

**Institutional utility.** The institution derives benefits from task success and incurs costs associated with worker effort, for example through compensation required to cover increased workload (Rosen, 1986). Let $b_I, \ell_I \geq 0$ denote the institutional benefit from task success and loss from task failure, respectively, and let $\xi \geq 0$ scale the worker's cost from the institutional perspective. The institutional utility under action $(d, s)$ is

$$U_I(d, s) := b_I \, p(d, s) - \ell_I \big(1 - p(d, s)\big) - \xi \, \text{cost}(d, s). \tag{3}$$

Institutional utility depends on the worker's action, which is not directly controlled by the institution.

**Worker quality.** We define *worker quality* as the institutional utility induced by the worker's optimal action,

$$Q(W) := U_I(d^\star, s^\star). \tag{4}$$

If institutional and worker utilities are aligned (e.g., $b_I = b_W$, $\ell_I = \ell_W$, and $\xi = 1$), then $Q(W) = U_W(d^\star, s^\star)$ and worker quality coincides with the worker's realized utility, which weakly improves under AI assistance. In general, however, $U_I \neq U_W$. Institutions employ multiple workers with heterogeneous objectives, so institutional utility cannot coincide with each worker's individual utility. This misalignment allows worker quality to decrease under AI assistance even when workers act rationally, consistent with empirical observations (Bogert et al., 2021; Passi & Vorvoreanu, 2022).

**Reliability and task success.** Task success under action $(d, s)$ arises through three mutually exclusive pathways: (i) direct worker execution, (ii) direct AI execution, and (iii) worker success after detecting and correcting an AI error. To compute the probability for each pathway, we define the reliability of the worker and the AI. Let $p_w, p_a \in [0, 1]$ denote the task success probabilities of the worker and the AI, respectively. The probabilities of direct worker and AI success are $(1 - d)p_w$ and $dp_a$. The quantities $p_w$ and $p_a$ may depend on task difficulty. For simplicity, we assume uniform task difficulty in the main text so that $p_w$ and $p_a$ are fixed, and relax this assumption in Section C.4. Worker correction after AI errors is governed by an *error detection function* $\phi : [0, 1] \to [0, 1]$, which maps verification effort $s$ to the probability of detecting an AI error. Conditional on detecting an error, the worker's subsequent task success is assumed to be independent of the AI's original output. Thus, the probability of success via correction is $d(1 - p_a)\phi(s)p_w$. Combining these cases, the task success probability induced by action $(d, s) \in [0, 1]^2$ is

$$p(d, s) := (1 - d)p_w + dp_a + d(1 - p_a)\phi(s)p_w. \tag{5}$$

When $p_a < p_w$, delegation without verification (i.e., $d > 0$ and $s = 0$) can reduce the overall task success probability. We impose the following regularity conditions on the error detection function $\phi(\cdot)$: (i) $\phi(0) = 0$, so no errors are detected without verification effort. (ii) $\phi$ is increasing, continuously differentiable, and strictly concave in $s$, implying diminishing returns to verification effort. These conditions capture the well-established speed–accuracy tradeoff and insights from random search theory in human cognition (Wickelgren, 1977; Heitz, 2014; Koopman, 1956).

A commonly used functional form is $\phi(s) := 1 - e^{-\alpha s}$, where $\alpha \geq 0$ represents the worker's verification reliability, namely the capacity to detect and correct AI errors after delegation. Larger values of $\alpha$ yield higher detection probabilities for any fixed $s > 0$, i.e., $\phi(s)$ is strictly increasing in $\alpha$. An alternative is the inverse-linear form $\phi(s) := 1 - \frac{1}{1+\alpha s}$, which is a first-order approximation to $1 - e^{-\alpha s}$ and is analytically convenient.

**Execution efficiency and cost.** We model the cost incurred

under action $(d, s)$ by aggregating execution, verification, and correction costs. Let $C_w, C_a \geq 0$ denote the execution costs of the worker and the AI, respectively. The expected execution cost is $(1 - d)C_w + dC_a$. In practice, $C_a$ is typically much smaller than $C_w$, and we often assume $C_a = 0$ for simplicity.

We parameterize the worker's execution cost $C_w$ by an efficiency parameter $\beta \geq 0$, where larger $\beta$ corresponds to higher efficiency and lower execution cost. For example, $C_w$ may scale linearly as $1 - \beta$ or inverse-linearly as $1/\beta$. Verification incurs an additional cost modeled by a *detection cost function* $C_v : [0, 1] \to \mathbb{R}_{\geq 0}$. We assume $C_v(0) = 0$ and that $C_v$ is increasing, continuously differentiable, and strictly convex, implying increasing marginal verification costs (Laffont & Martimort, 2002). Intuitively, early verification checks salient or easily detectable errors, while deeper verification requires increasingly costly activities such as independent recomputation, external cross-checking, or domain-expert consultation. Examples include a linear cost $C_v(s) \propto s$ or a convex cost such as $C_v(s) = s + \frac{s^2}{2}$. Since verification occurs only when the task is delegated to AI, the expected verification cost under action $(d, s)$ is $d\, C_v(s)$. If an AI error is detected, the worker redoes the task at cost $C_w$, yielding an expected correction cost of $d(1 - p_a)\phi(s)C_w$. The total cost under action $(d, s)$ is

$$
\begin{aligned}
\text{cost}(d, s) := (1 - d)C_w \\
+ d\left(C_a + C_v(s) + (1 - p_a)\phi(s)C_w\right).
\end{aligned}
\tag{6}
$$

Substituting (5) and (6) into (1) and (3) yields $U_W(d, s) = f_W(s)d + g_W$ and $U_I(d, s) = f_I(s)d + g_I$, both linear in the delegation level $d$. Here, $g_W$ and $g_I$ denote the worker's and institution's no-AI baseline utilities; in particular, $g_I$ equals the pre-AI worker quality $Q_0(W)$. We restrict to the regime $g_W \geq 0$, ensuring nonnegative pre-AI worker utility. The terms $f_W(s)d$ and $f_I(s)d$ capture the incremental utilities from delegation with verification effort $s$. Expressions for $f_W(s), g_W, f_I(s)$, and $g_I$ are given in Section A.1. Because worker and institutional objectives need not align, it is possible that $f_W(s) > 0$ while $f_I(s) < 0$, so delegation benefits the worker but harms the institution.

*Remark* 2.1 (**Structural properties of delegation with verification**). The discontinuities identified in subsequent sections do not depend on specific functional forms, but arise from structural features of utility-driven delegation with costly verification. Suppose that (i) $\phi(s)$ is increasing and concave, (ii) $C_v(s)$ is increasing and convex, and (iii) utilities are affine in the delegation probability $d$ for fixed $s$. Then $U_W(d, s) = f_W(s)d + g_W$, implying $d^\star(s) \in \{0, 1\}$ whenever $f_W(s) \neq 0$. As a result, small changes in worker or AI ability that flip the sign of $f_W(s)$ induce abrupt transitions between no delegation, verified delegation, and pure delegation. These phase transitions are therefore generic to delegation pipelines with costly verification.

In the remainder of the paper, we analyze how the optimal action $(d^\star, s^\star)$ and resulting quality $Q(W)$ vary with verification reliability $\alpha$ and execution efficiency $\beta$. By contrast, in the pre-AI baseline only execution efficiency $\beta$ matters, as verification is unnecessary.

## 3. Theoretical Results

This section presents our main theoretical results on how AI assistance reshapes worker behavior and institutional worker quality. Throughout, we fix the task profile (benefit parameters $b_W, b_I$, loss parameters $\ell_W, \ell_I$, and discount factor $\xi$), AI characteristics (task success probability $p_a$ and execution cost $C_a$), and baseline worker characteristics (task success probability $p_w$ and verification cost function $C_v(\cdot)$). Under this setup, the worker's optimal action $(d^\star, s^\star)$ and resulting worker quality $Q(W)$ depend only on verification reliability $\alpha$ and execution efficiency $\beta$, which parameterize the detection function $\phi(\cdot)$ and execution cost $C_w$.

We first show that the optimal worker action is nonlinear and discontinuous in $(\alpha, \beta)$ (Theorem 3.1). We then analyze how AI assistance alters worker quality relative to the pre-AI baseline: Theorem 3.5 characterizes when AI improves worker quality, while Theorem 3.6 identifies which workers are upgraded or downgraded, highlighting the central role of verification ability.

**Characterization of worker action.** To compute the verification decision, we define the *verification surplus* function

$$
\Phi(s; \alpha, \beta) := (1 - p_a)\left((b_W + \ell_W)p_w - C_w\right)\phi(s) - C_v(s),
$$

which captures the marginal benefit of detecting AI errors net of verification cost. This function corresponds to the $s$-dependent component of $f_W(s)$. Let

$$
s^\dagger(\alpha, \beta) \in \arg\max_{s \in [0, 1]} \Phi(s; \alpha, \beta)
$$

denote the optimal verification effort, conditional on verification being employed. Since $f_W(s) = \Phi(s; \alpha, \beta) + $ const, where the constant term does not depend on $s$. By the definition of $\Phi$, this is equivalent to $s^\dagger(\alpha, \beta) \in \arg\max_{s \in [0, 1]} f_W(s)$. We also define the *manual-to-delegation gain*

$$
\Delta(\beta) := U_W(1, 0) - U_W(0, 0) = f_W(0),
\tag{7}
$$

which represents the worker's net utility change when switching from manual work to pure delegation (i.e., delegating execution to AI without verification). The last equality follows from the decomposition $U_W(d, s) = f_W(s)\, d + g_W$. Since no verification effort is exerted, we have $\phi(0) = 0$ and $C_v(0) = 0$ for any $\alpha$, and hence $\Delta$ depends only on the execution efficiency $\beta$. Below we characterize the utility-driven worker action.

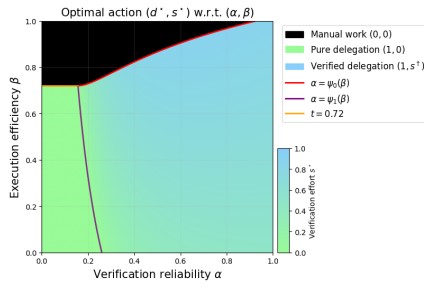

*(a)* Worker action in Theorem 3.1

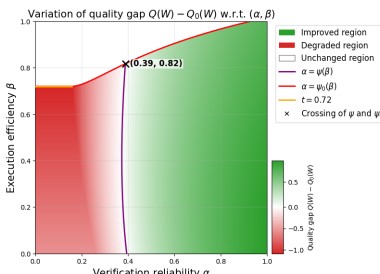

*(b)* Quality gap in Theorem 3.5

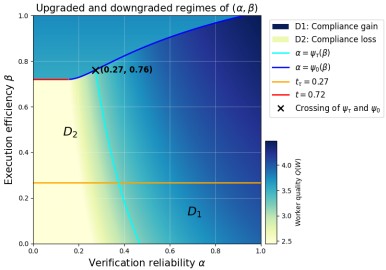

*(c)* Quality grading in Theorem 3.6

*Figure 1.* Plots illustrating the worker action and quality regimes characterized in Theorems 3.1, 3.5, and 3.6 as functions of abilities $(\alpha, \beta)$, under the default parameter setting $(b_W, \ell_W, b_I, \ell_I, \xi, \tau, p_a, C_a, p_w) = (8, 6, 14, 12, 0.3, 6.4, 0.65, 0, 0.75)$, and the functional choices $C_v(s) = s$, $C_w(\beta) = 5(1 - \beta)$, and $\phi(s; \alpha) = 1 - \frac{1}{1 + 2\alpha s}$. Discussion on the choice of parameters and functions, and closed-form expressions for key quantities—including $(d^\star, s^\star)$ and the boundaries $\psi_0, \psi_1, \psi,$ and $\psi_\tau$—are provided in Section A.10.

**Theorem 3.1 (Worker optimal action).** *Fix the task profile, AI characteristics, and baseline worker characteristics. Let $t \geq 0$ be a threshold such that $\Delta(t) = 0$. There exist continuous functions $\psi_0 : [t, \infty) \to \mathbb{R}_{\geq 0}$, monotonically increasing, and $\psi_1 : [0, t] \to \mathbb{R}_{\geq 0}$, monotonically decreasing, such that, for all $\beta$ in their respective domains,*

$$f_W\big(s^\dagger(\psi_0(\beta), \beta)\big) = 0 \quad and \quad \partial_s \Phi\big(0; \psi_1(\beta), \beta\big) = 0.$$

*Then the optimal action $(d^\star, s^\star)$ takes the following form:*

- **(Manual work)** $(0, 0)$ *if $(\beta \geq t)$ and $(\alpha < \psi_0(\beta))$;*
- **(Pure delegation)** $(1, 0)$ *if $(\beta < t)$ and $(\alpha < \psi_1(\beta))$;*
- **(Verified delegation)** $(1, s^\dagger(\alpha, \beta))$ *otherwise.*

This result shows that the worker's optimal action lies in one of three regimes: manual work $(0, 0)$, pure delegation $(1, 0)$, and verified delegation $(1, s^\dagger(\alpha, \beta))$. The characterization therefore partitions the ability space $(\alpha, \beta)$ into distinct workflow regimes, with regime boundaries determined by the worker's delegation and verification incentives. In particular, the optimal delegation decision $d^\star$ is binary, leading to sharp regime shifts. This discreteness follows from the fact that worker utility $U_W(d, s)$ is affine in the delegation variable $d$, highlighting the structural source of abrupt action transitions under AI assistance (Figure 1a). For example, as $\beta$ decreases past a threshold $t$, the worker may abruptly switch from manual work to pure delegation. Similarly, as verification reliability $\alpha$ crosses a threshold $\psi_0(\beta)$, the optimal verification effort $s^\star$ can jump discontinuously from 0 to $s^\dagger(\alpha, \beta)$. These phase transitions provide an explanation for empirical findings that accountability failures can emerge abruptly rather than gradually in AI-assisted workflows (Zhang & Deniz, 2024).

The threshold condition $\Delta(t) = 0$ characterizes indifference between manual work and pure delegation. The condition $f_W\big(s^\dagger(\psi_0(\beta), \beta)\big) = 0$ implies that, at $(\alpha, \beta) = (\psi_0(\beta), \beta)$, the worker is indifferent between manual work and verified delegation, so $\psi_0$ separates these regimes. Fi-

nally, the condition $\partial_s \Phi\big(0; \psi_1(\beta), \beta\big) = 0$ implies that verification yields zero marginal surplus at $s = 0$, so $\psi_1$ separates pure and verified delegation. When $\alpha$ is sufficiently large, the worker enters the verified-delegation regime, underscoring the role of verification ability in preventing over-delegation. Moreover, the monotonicity of the separatrices $\psi_0$ and $\psi_1$ implies that the optimal delegation decision $d^\star$ is non-increasing in execution efficiency $\beta$: higher execution efficiency makes manual work more attractive, reducing reliance on AI.

**Worker quality under AI assistance.** Given the worker's optimal action $(d^\star, s^\star)$, we can compute the resulting worker quality $Q(W)$. A central question is whether $Q(W)$ exceeds the no-AI baseline quality $Q_0(W)$. This question is practically relevant for workers, who seek to remain qualified and avoid job loss. It is equally important for institutions, for whom worker quality directly determines productivity, liability, and risk. From the institutional perspective, a key concern is whether workers are upgraded or downgraded under AI assistance, as this directly affects productivity. Let $\tau \geq 0$ denote a qualification threshold for worker quality. Formally, we ask which workers are *upgraded*—moving from $Q_0(W) < \tau$ to $Q(W) \geq \tau$—and which are *downgraded*—moving from $Q_0(W) \geq \tau$ to $Q(W) < \tau$.

To address these questions, we introduce technical assumptions. The first ensures that the institution places higher marginal value on task success than the worker does.

**Assumption 3.2 (Institutional dominance).** Assume that $(b_I + \ell_I) > \xi (b_W + \ell_W)$.

This assumption is typically reasonable in practice: institutions often have larger benefits from success and larger losses from failure than individual workers (i.e., $b_I > b_W$ and $\ell_I > \ell_W$), while the institution may bear only a discounted fraction of the worker's private cost (i.e., $\xi < 1$).

A second assumption ensures that, under the worker's optimal verification choice, higher verification reliability $\alpha$

leads to higher detection probability.

**Assumption 3.3** (**Monotonicity of detection**). *Assume that for any $\alpha, \beta \geq 0$, $\partial_\alpha \phi\big(s^\dagger(\alpha, \beta)\big) \geq 0$.*

This assumption is intuitive: more reliable verifiers should detect more under optimal behavior. We provide concrete examples validating this condition in Section A.2. We first study whether AI assistance improves worker quality relative to the no-AI baseline, i.e., whether $Q(W) - Q_0(W) > 0$ or not. Recall that $Q_0(W) = g_I$, which immediately implies the following claim.

**Proposition 3.4** (**Worker quality difference**). $Q(W) - Q_0(W) = f_I(s^\star) d^\star$.

Consequently, the question reduces to determining the sign of $f_I(s^\star) d^\star$, which leads to the following theorem.

**Theorem 3.5** (**Worker quality improvement v.s. no AI**). *Fix the task profile, AI characteristics, and baseline worker characteristics. Let $t$ and $\psi_0(\cdot)$ be as given in Theorem 3.1. Under Assumptions 3.2 and 3.3, there exists a continuous function $\psi : \mathbb{R}_{\geq 0} \to \mathbb{R}_{\geq 0}$ such that*

$$(f_I(s^\star) = 0) \wedge (d^\star = 1) \text{ for any pair } (\alpha, \beta) = (\psi(\beta), \beta).$$

*Moreover,*

- (**Improved**) $Q(W) > Q_0(W)$ *holds iff* $\alpha > \psi(\beta)$;
- (**Unchanged**) $Q(W) = Q_0(W)$ *holds iff* $\big((\beta \geq t) \wedge (\alpha < \psi_0(\beta))\big)$ *or* $\alpha = \psi(\beta)$;
- (**Degraded**) $Q(W) < Q_0(W)$ *holds otherwise.*

Thus, AI assistance reshapes worker quality in heterogeneous ways. The regime in which quality remains unchanged largely coincides with the manual-work regime $((d^\star, s^\star) = (0, 0))$. Beyond this, worker quality improves with high verification reliability $\alpha$ but can degrade with low $\alpha$, as over-delegation reduces task success. Moreover, a higher execution efficiency $\beta$ does not necessarily translate into improved worker quality under AI assistance since delegation behavior can weaken the induced utility gain.

Figure 1b visualizes the three regimes governing how worker quality varies under AI assistance in Theorem 3.5. Notably, there is a specific region in which quality is degraded, located below the intersection point $(\alpha, \beta) = (0.39, 0.82)$ of the separatrices $\psi$ and $\psi_0$. This region highlights a countervailing effect: increasing verification reliability can induce workers to switch from manual work to delegation even when reliability remains insufficient to make delegation quality-improving, thereby triggering over-delegation and reducing quality.

Moreover, we have $t = 1 - \frac{C_a + (b_W + \ell_W)(p_w - p_a)}{5}$ (see (16)), which is monotonically increasing in AI ability (i.e., higher $p_a$ or lower $C_a$). Consequently, more capable AI enlarges

the quality-degradation regime when $\alpha$ is close to $0$. This reveals a second countervailing effect: improved AI capability can induce workers with low verification reliability to over-delegate, thereby reducing quality; see also Figure 4a.

Next, we characterize the conditions under which workers are upgraded or downgraded under AI assistance.

**Theorem 3.6** (**Worker upgraded v.s. downgraded**). *Fix the task profile, AI characteristics, and baseline worker characteristics. Fix a grading threshold $\tau \geq 0$. Let $t_\tau \geq 0$ be the solution to $Q_0(W; \beta) = \tau$. Under Assumptions 3.2 and 3.3, there exists a monotonically decreasing function $\psi_\tau : \mathbb{R}_{\geq 0} \to \mathbb{R}_{\geq 0}$ such that $Q(W) = \tau$ for the ability pair $(\alpha, \beta) = (\psi_\tau(\beta), \beta)$. Then*

- (**Compliance gain**) $(Q(W) > \tau) \wedge (Q_0(W) < \tau)$ *holds iff* $\big(\beta < \min\{t, t_\tau\}\big) \wedge \big(\alpha > \psi_\tau(\beta)\big)$ *or* $\big(t \leq \beta < t_\tau\big) \wedge \big(\alpha > \max\{\psi_0(\beta), \psi_\tau(\beta)\}\big)$.
- (**Compliance loss**) $(Q(W) < \tau) \wedge (Q_0(W) > \tau)$ *holds iff* $\big(t_\tau < \beta \leq t\big) \wedge \big(\alpha < \psi_\tau(\beta)\big)$ *or* $\big(\beta > \max\{t, t_\tau\}\big) \wedge \big(\psi_0(\beta) < \alpha < \psi_\tau(\beta)\big)$.

This result identifies both a *compliance gain* regime and a *compliance loss* regime, separated by a boundary that depends on verification reliability $\alpha$. In particular, high $\alpha$ enables workers with low execution efficiency ($\beta < t_\tau$) to upgrade, while low $\alpha$ can cause workers with high execution efficiency ($\beta > t_\tau$) to be downgraded. The monotonicity of the separatrix $\psi_\tau$ indicates that workers with lower execution efficiency require higher verification reliability to remain qualified.

Together with Theorem 3.5, this yields a *verification amplification* phenomenon: AI access can disproportionately benefit workers with strong evaluative capabilities, while disadvantaging others through rational over-delegation. Notably, these effects arise endogenously from rational optimization, without invoking behavioral bias or miscalibration.

Figure 1c visualizes the upgraded and downgraded regimes characterized in Theorem 3.6. When $\beta < t$, worker quality $Q(W)$ increases more rapidly with $\alpha$ at low values of $\alpha$, revealing diminishing returns to verification upskilling. This highlights that verification upskilling is most valuable for workers with low initial reliability.

In Section 4, we study multiple interventions and model extensions. We first analyze how worker-side and institutional interventions reshape the action, quality, and compliance regimes. We then relax several baseline assumptions, including task difficulty, belief calibration, and correction workflows. Across these variants, the core qualitative mechanism remains stable: AI assistance improves outcomes only when delegation is paired with sufficiently reliable verification, whereas rational delegation can still generate quality loss when verification is weak.

## 3.1. Overview of the proofs

We summarize the main proof ideas; full proofs appear in Section A. All arguments rely on monotonicity properties of a small number of quantities with respect to verification reliability $\alpha$ and execution efficiency $\beta$, despite the fact that optimal verification effort depends implicitly on both parameters. These relationships are summarized in Table 2.

**To Theorem 3.1.** We first characterize the structure of the worker's optimal workflow. Recall that $U_W(d, s) = f_W(s)\,d + g_W$ is affine in the delegation variable $d$, with all nontrivial dependence on verification captured by $f_W(s)$. Under strict concavity of $\phi$ and strict convexity of $C_v$, $f_W(s)$ is strictly concave, implying a unique optimal verification effort $s^\dagger(\alpha, \beta) \in \arg\max_{s\in[0,1]} f_W(s)$. The optimal delegation decision is therefore binary:

$$d^\star = \mathbb{I}\big[f_W\big(s^\dagger\big) \geq 0\big],$$

with the boundary between delegation and manual work given by $f_W\big(s^\dagger\big) = 0$.

To locate this boundary in ability space, we study how $f_W\big(s^\dagger(\alpha, \beta)\big)$ varies with $\alpha$ and $\beta$.

**Lemma 3.7** (**Effects of $\alpha$ and $\beta$ on $f_W$**). $f_W\big(s^\dagger(\alpha, \beta)\big)$ *is non-decreasing in $\alpha$ and non-increasing in $\beta$.*

When $s^\dagger = 0$, the condition $f_W(s^\dagger) = 0$ coincides with the manual–delegation indifference condition $\Delta(\beta) = 0$, yielding the threshold $t$ separating manual work from pure delegation. When $s^\dagger > 0$, the same condition defines the separatrix $\psi_0$ between manual work and verified delegation. The monotonicity of $\psi_0$ follows directly from Lemma 3.7.

The remaining boundary $\psi_1$, separating pure from verified delegation, is determined by whether verification yields positive marginal surplus at $s = 0$.

**Lemma 3.8** (**Effects of $\alpha$ and $\beta$ on $\Phi$**). $\partial_s\Phi(0; \alpha, \beta)$ *is non-decreasing in $\alpha$ and non-decreasing in $\beta$.*

Since $\psi_1$ is defined by $\partial_s\Phi(0; \alpha, \beta) = 0$, this lemma determines its geometry. Together, Lemmas 3.7 and 3.8 characterize the three workflow regimes and explain the origin of discontinuous transitions.

**To Theorem 3.5.** Theorem 3.5 compares worker quality with and without AI assistance. By Proposition 3.4, the quality gap reduces to the sign of $f_I(s^\star)d^\star$, so the key question is how this quantity varies with verification reliability.

**Lemma 3.9** (**Effect of $\alpha$ on $f_I(s^\star)d^\star$**). $d^\star$ *is non-decreasing in $\alpha$; and for ability pairs $(\alpha, \beta)$ with $d^\star = 1$, $f_I(s^\star)$ is non-decreasing in $\alpha$.*

Higher $\alpha$ weakly increases the likelihood of delegation and, conditional on delegation, increases the institutional value of verification. Assumptions 3.2 and 3.3 ensure that detection improvements translate into institutional value at least as fast as into worker utility, yielding a separatrix $\psi$ such that worker quality improves exactly when $\alpha > \psi(\beta)$.

**To Theorem 3.6.** Finally, we characterize which workers are upgraded or downgraded relative to a qualification threshold $\tau$. This requires understanding how worker quality varies jointly with $\alpha$ and $\beta$ in the delegation regime.

**Lemma 3.10** (**Effects of $\alpha, \beta$ on $Q$**). *For ability pairs with $d^\star = 1$, worker quality $Q(W)$ is non-decreasing in $\alpha$ and non-decreasing in $\beta$.*

This lemma determines the geometry of the level sets $Q(W) = \tau$, yielding the separatrix $\psi_\tau$ that divides compliance gain from compliance loss. Combined with Theorem 3.1, this shows how AI assistance can upgrade workers with strong verification ability while downgrading others through rational over-delegation.

## 4. Interventions and Model Extensions

The model in Section 2 studies delegation and verification in the $(\alpha, \beta)$ ability space under simplifying assumptions such as fixed AI capability, calibrated beliefs, homogeneous tasks, single-round verification, full re-execution, binary outcomes, and regular verification technologies. We then summarize interventions and extensions that modify these assumptions. Across these variants, the same qualitative structure persists: actions remain organized into manual work, pure delegation, and verified delegation, and the model continues to exhibit compliance gain/loss and rational over-delegation. Full details appear in Sections B and C.

**Interventions.** *Worker-side upskilling (Section B.1).* Workers may meet the institutional standard $\tau$ by investing in verification reliability and/or execution efficiency. Let $h_1$ and $h_2$ denote the costs of increasing $\alpha$ and $\beta$, respectively. The least-cost worker-side intervention is given by

$$\min_{D_\alpha, D_\beta \geq 0} h_1(D_\alpha) + h_2(D_\beta) \text{ s.t. } Q(\alpha + D_\alpha, \beta + D_\beta) \geq \tau.$$

Figure 3 shows that such adjustment often favors increasing $\alpha$, highlighting verification reliability's importance.

*Deploying a more capable AI system (Section B.2).* The institution may consider stronger AI to improve worker quality, represented by $p_a \mapsto p_a + D_p, D_p \geq 0$. Figure 4a shows that a more capable AI system can have a non-monotonic effect on worker quality.

*Incentive reshaping and equilibrium design (Section B.2).* To strengthen substantive accountability, the institution may transfer a reward $D_b \geq 0$ from its own benefit $b_I$ to the worker's benefit $b_W$. Figures 4b and 4c compare uniform and worker-specific choices of $D_b$: fixed transfers have heterogeneous effects, while optimized transfers can act as an alignment mechanism when incentive alignment enters

the institutional objective.

*Rewarding observable verification (Section B.2).* If verification is observable, the institution can reward verification effort through a process-based reward $R(s)$, which modifies the worker and institutional utilities as follows:

$$U_W^R(d,s) = U_W(d,s) + dR(s), \ U_I^R(d,s) = U_I(d,s) - dR(s).$$

Figure 5 shows that verification rewards expand verified delegation and shrink quality-degradation regions, with ability-dependent effects.

**Model extensions.** *Partial delegation (Section C.1).* We relax binary reliance by adding a convex cost of delegation intensity: $-\delta d^2$ to $U_W$. Figure 6 shows that this creates an interior partial-delegation region $d^\star \in (0,1)$ and smooths the delegation transition, while preserving the main action-regime, compliance, and over-delegation patterns.

*Time and cognitive resource constraints (Section C.2).* We restrict feasible actions by a time or cognitive budget:

$$\mathcal{A}_{\tau_c} = \{(d,s) \in [0,1]^2 : (1-d)C_w + dC_v(s) \le \tau_c\}.$$

Figure 7 shows that such constraints can make manual work or intensive verification infeasible, shifting workers toward delegation or lower verification.

*Belief miscalibration (Section C.3).* We allow the worker to optimize under a subjective AI success probability $\widehat{p}_a$, while institutional quality is evaluated under the true $p_a$. Figure 8 shows that overestimation ($\widehat{p}_a > p_a$) intensifies over-delegation and expands quality-degradation and compliance-loss regions, whereas underestimation ($\widehat{p}_a < p_a$) expands manual work and dampens both gains and losses.

*Heterogeneous task difficulty (Section C.4).* We allow success probabilities and costs to vary with task difficulty $h$, and define worker quality by averaging $Q(W;h)$ over the task distribution: $Q(W) = \int Q(W;h)\, d\mu(h)$. Figure 9 shows that task heterogeneity reshapes the quality regions but preserves the main improvement/degradation and compliance gain/loss patterns.

*Partial re-execution (Section C.5).* We allow detected AI errors to be corrected through partial rather than full re-execution, replacing the correction cost $(1-p_a)\phi(s)C_w$ by $(1-p_a)\phi(s)\kappa C_w$. Figure 10 shows that this lowers the cost of verified delegation but leaves the delegation–verification structure and qualitative regions essentially unchanged.

*Multi-round interaction (Section C.6).* We reduce a $T$-round interaction to an effective one-shot verification problem. Given total verification effort $s$, the multi-round process induces an effective detection frontier $\bar{\phi}_T(s)$ and an effective verification cost $\bar{C}_{v,T}(s)$. Substituting these objects for $\phi(s)$ and $C_v(s)$ preserves the same reduced-form structure as the single-round model, showing that the qualitative mechanism does not depend on single-round verification.

*Continuous output quality (Section C.7).* Replacing binary success probabilities with expected qualities, $p_w \mapsto \mathbb{E}[Y_w]$ and $p_a \mapsto \mathbb{E}[Y_a]$, preserves the same reduced-form structure, so the model extends directly to continuous output quality.

*Relaxed regularity conditions (Section C.8).* We relax the concavity of the detection function and the convexity of the verification cost. Figure 11 shows that verification choices may become less smooth, but the main qualitative conclusions remain unchanged.

## 5. From Data to Model Parameters

While our analysis is structural rather than empirical, the model is designed to interface with observational data from AI-assisted workflows. The exercise in this section should therefore be interpreted as a data-to-parameter mapping rather than an empirical validation of the model's causal claims. We outline how commonly available institutional data can be mapped to the model's parameters, with full implementation details deferred to Section D.

**Observational inputs.** We use the Collab-CXR dataset (Agarwal et al., 2023), which contains diagnostic data for 324 patient cases from 33 clinicians. For each clinician–case pair, the dataset records predictions over 14 pathologies in three settings: clinician alone, AI-alone, and clinician with access to the AI's output. The dataset also reports time spent per case in the clinician-alone and clinician-with-AI conditions. Expert diagnoses from 10 radiologists provide a reference for evaluating correctness, and a washout period of at least two weeks is enforced between clinician-alone and clinician-with-AI sessions.

**Mapping observables to model quantities.** From these data, we map each clinician to the model's core quantities: task success probabilities $p_w$ and $p_a$, success probability under AI assistance $p(1,s^\dagger)$, execution costs $C_w$ and $C_a$, and the assisted cost $\text{cost}(1,s^\dagger)$. We illustrate this mapping for clinician vn_exp_65341958.

*Deriving $p_w$, $p_a$, and $p(1,s^\dagger)$.* Task success is defined as matching expert labels on all 14 pathologies. Under this criterion, we obtain success indicators for $n = 38$ cases in the clinician-alone, AI-alone, and clinician-with-AI conditions. Averaging yields $p_w = 0.447$, $p_a = 0.368$, and $p(1,s^\dagger) = 0.395$. In the model, the clinician-with-AI condition corresponds to $d = 1$, with the clinician exerting optimal verification effort $s^\dagger$.

*Deriving $C_w$, $C_a$, and $\text{cost}(1,s^\dagger)$.* We assume worker cost is proportional to task completion time. Since AI execution time is negligible, we set $C_a = 0$. Average completion times over the $n$ cases yield $C_w = 172.5$ and $\text{cost}(1,s^\dagger) = 116.8$.

We next infer the clinician's ability parameters $(\alpha, \beta)$.

*Deriving $\phi(s^\dagger)$.* By (5), $\phi(s^\dagger) = \frac{p(1,s^\dagger)-p_a}{(1-p_a)p_w} = 0.093$.

*Deriving $C_v(s^\dagger)$.* By (6) with $C_a = 0$, $C_v(s^\dagger) = \text{cost}(1, s^\dagger) - (1 - p_a)\phi(s^\dagger)C_w = 106.7$.

*Choice of ability functions.* We model cost through time expenditure, taking linear verification cost and normalizing verification effort and execution efficiency via $C_v(s) = T_v^{\max}s$ and $C_w(\beta) = T_w^{\max}(1 - \beta)$, where $T_v^{\max} = 118.1$ and $T_w^{\max} = 262.3$ are the maximum observed verification and execution times. These normalizations ensure $s^\dagger, \beta \in [0, 1]$. We use $\phi(s; \alpha) = 1 - e^{-\alpha s}$ (Heitz, 2014).

*Deriving $\beta$.* From $C_w$, we obtain $\beta = 1 - \frac{C_w}{T_w^{\max}} = 0.342$.

*Deriving $\alpha$.* From $C_v(s) = T_v^{\max}s$, we obtain $s^\dagger = 0.903$. Using $\phi(s^\dagger; \alpha)$ and the estimated $\phi(s^\dagger)$ yields $\alpha = -\frac{\ln(1 - \phi(s^\dagger))}{s^\dagger} = 0.108$.

**Identifying regime membership.** Given $(\alpha, \beta)$, the model predicts the clinician's optimal action and resulting quality.

*Identifying the action regime.* By rationality, $s^\dagger$ satisfies $\partial_s \Phi(s^\dagger; \alpha, \beta) = 0$, yielding

$$b_W + \ell_W = \frac{T_v^{\max} + (1 - p_a)\alpha e^{-\alpha s^\dagger}C_w}{(1 - p_a)\alpha e^{-\alpha s^\dagger}p_w} = 4646.0.$$

This implies $f_W(s^\dagger) = -189.26 < 0$, so $(d^\star, s^\star) = (0, 0)$. The clinician therefore lies in the manual-work regime of Theorem 3.1.

*Identifying the quality regime.* Setting $b_I = 2787.6$, $\ell_I = 1858.4$, and $\xi = 0.5$ (Assumption 3.2) yields $Q(W) = Q_0(W) = 133.77$. Thus, the clinician lies in the quality-unchanged regime of Theorem 3.5, and is neither upgraded nor downgraded in Theorem 3.6.

**Interpreting interventions.** Suppose the institution raises the qualification threshold to $\tau = 150$. As discussed in Section 3, the model can evaluate interventions such as up-skilling worker abilities (increasing $\alpha$ or $\beta$) or deploying more capable AI (increasing $p_a$). To achieve $Q(W) \geq \tau$, the institution may consider: (i) increasing $\alpha$ from 0.108 to 0.335; (ii) increasing $\beta$ from 0.342 to 0.479; or (iii) increasing $p_a$ from 0.368 to 0.432. This numerical analysis can guide for institutions when designing intervention policies.

**Scope and limitations.** This calibration illustrates applicability rather than identification. The model does not estimate causal effects or capture dynamic learning or organizational adaptation. Parameters such as $\alpha$ and $\beta$ may be interpreted at the role or population level, depending on the setting.

# 6. Conclusions, Limitations, and Future Work

This paper develops a formal framework for understanding how AI assistance reshapes institutional worker quality through endogenous delegation and verification decisions. By modeling worker behavior as the solution to a rational optimization problem and evaluating outcomes through an institution-centered notion of quality, we identify a structural mechanism by which AI can induce sharp, nonlinear changes in behavior and outcomes.

We show that AI assistance transforms the mapping from worker ability to workflow choice. Small differences in verification reliability can trigger abrupt regime shifts between manual work, verified delegation, and pure delegation. These phase transitions arise from structural properties of delegation with costly verification and persist under general assumptions on detection and cost functions.

As a result, AI assistance reshapes worker quality heterogeneously. Workers with strong verification ability may experience *compliance gains*, becoming institutionally qualified despite lower execution efficiency, while workers with weaker verification ability may over-delegate and incur *compliance loss*, even when baseline task success improves. These effects arise endogenously from rational optimization, without invoking behavioral bias or miscalibration.

Taken together, our results suggest that in AI-assisted workflows, the ability to evaluate and verify AI outputs becomes a central determinant of institutional outcomes. AI thus functions not only as a productivity-enhancing tool, but as a structural filter that amplifies differences in verification ability. At the organizational level, this suggests that AI deployment should be accompanied by explicit attention to verification protocols, correction procedures, and accountability arrangements.

Our framework abstracts away from several real-world considerations to isolate this core mechanism. We study a single worker performing a single task, treat verification reliability as exogenous, and do not model dynamic effects such as learning or deskilling. Institutional evaluation is outcome-based and does not condition on workflow observability. Moreover, our analysis is structural rather than empirical and does not estimate effect sizes in specific domains.

These limitations suggest several directions for future work. Promising extensions include dynamic models in which execution efficiency and verification reliability evolve over time, analyses of institutional interventions such as verification requirements or incentive schemes, and multi-worker or team-based settings with distributed oversight. Empirical studies mapping real-world tasks and verification practices to model parameters could further enable quantitative assessment of compliance gain, compliance loss, and verification amplification in practice.

## Impact statement

This work develops a formal framework for analyzing human–AI collaboration in institutional settings where delegation and verification decisions are shaped by misaligned incentives. By isolating structural mechanisms that govern oversight behavior, the paper contributes to a clearer understanding of how AI systems may alter human roles in high-stakes decision-making processes.

The analysis highlights a potential societal risk in AI-assisted workflows: as AI capability improves, rational agents may reduce verification effort, which can degrade institutional outcomes even when average task performance increases. This mechanism is particularly relevant for domains such as healthcare, law, and safety-critical decision support, where rare but consequential errors must be detected through human oversight. If left unaddressed, such dynamics may contribute to uneven accountability or hidden failure modes in automated decision pipelines.

The results also suggest potential fairness implications. Because verification reliability plays a central role in determining outcomes under AI assistance, the benefits of AI may accrue disproportionately to individuals or groups with stronger evaluative capabilities, while others may experience performance degradation despite rational behavior. This raises questions about how institutions evaluate workers and distribute responsibility in AI-mediated settings.

From a safety and system-design perspective, the framework suggests that improvements in AI accuracy alone are insufficient to guarantee robust human–AI collaboration. Within the scope of the model, sustaining effective oversight requires institutional mechanisms that explicitly value verification effort, such as structured review protocols or incentive schemes that reward error detection rather than output accuracy alone.

This paper is theoretical in nature and does not estimate causal effects, model adversarial behavior, or assess regulatory compliance directly. Further empirical and policy-oriented work is needed to evaluate how the identified mechanisms interact with real-world constraints. The framework is intended to support more informed design and evaluation of AI-assisted workflows, rather than to advocate for increased or decreased automation in any specific domain.

## Acknowledgment

LH is supported in part by Fundamental and Interdisciplinary Disciplines Breakthrough Plan of the Ministry of Education of China (No. JYB2025XDXM118), NSFC Grant No. 625707396. NKV was supported in part by NSF Grant CCF-2112665.

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

*Table 1.* Notations used in this paper

| Symbol | Domain | Description |
|---|---|---|
| $d$ | $[0,1]$ | Delegation rate (fraction of task delegated to AI) |
| $s$ | $[0,1]$ | Verification (scrutiny) effort |
| $p(d,s)$ | $[0,1]^2 \to [0,1]$ | Task success probability induced by $(d,s)$ |
| $\mathrm{cost}(d,s)$ | $[0,1]^2 \to \mathbb{R}_{\geq 0}$ | Total cost induced by $(d,s)$ |
| $d^\star$ | $[0,1]$ | Worker-optimal delegation choice |
| $s^\star$ | $[0,1]$ | Worker-optimal verification choice |
| $b_W$ | $\mathbb{R}_{\geq 0}$ | Worker valuation per unit success probability |
| $b_I$ | $\mathbb{R}_{\geq 0}$ | Institutional valuation per unit success probability |
| $\ell_W$ | $\mathbb{R}_{\geq 0}$ | Worker loss per unit failure probability |
| $\ell_I$ | $\mathbb{R}_{\geq 0}$ | Institutional loss per unit failure probability |
| $\xi$ | $\mathbb{R}_{\geq 0}$ | Institutional cost discount factor |
| $p_w$ | $[0,1]$ | Baseline task success probability of worker |
| $p_a$ | $[0,1]$ | Baseline task success probability of AI |
| $\phi(s)$ | $[0,1] \to [0,1]$ | Detection probability as a function of verification effort $s$ |
| $C_w$ | $\mathbb{R}_{\geq 0}$ | Worker task execution cost |
| $C_a$ | $\mathbb{R}_{\geq 0}$ | AI task execution cost |
| $C_v(s)$ | $[0,1] \to \mathbb{R}_{\geq 0}$ | Verification cost as a function of scrutiny $s$ |
| $\alpha$ | $\mathbb{R}_{\geq 0}$ | Detection sensitivity parameter in $\phi$ |
| $\beta$ | $\mathbb{R}_{\geq 0}$ | Worker task efficiency parameter |
| $U_W(d,s)$ | $[0,1]^2 \to \mathbb{R}$ | Worker utility |
| $U_I(d,s)$ | $[0,1]^2 \to \mathbb{R}$ | Institutional utility |
| $Q(\alpha,\beta)$ | $\mathbb{R}_{\geq 0}^2 \to \mathbb{R}$ | Worker quality with AI |
| $Q_0(\alpha,\beta)$ | $\mathbb{R}_{\geq 0}^2 \to \mathbb{R}$ | Worker quality without AI |
| $\Delta_Q(\alpha,\beta)$ | $\mathbb{R}_{\geq 0}^2 \to \mathbb{R}$ | Quality gap induced by AI assistance |
| $\tau$ | $\mathbb{R}_{\geq 0}$ | Qualification threshold |

*Table 2.* Summary of monotonicity properties of key quantities with respect to the ability parameters $\alpha$ and $\beta$. Here $\uparrow$ denotes increasing, $\downarrow$ denotes decreasing, / denotes no clear monotonic pattern, and $\times$ denotes no dependence.

| | $s^\dagger$ | $d^\star$ | $s^\star$ | $Q_0$ | $Q\mid_{d^\star=1}$ | $Q\mid_{d^\star=1}-Q_0$ | $\psi_0(\beta)$ | $\psi_1(\beta)$ | $\psi(\beta)$ | $\psi_\tau(\beta)$ | $f_W(s^\dagger)$ | $\partial_s\Phi(0)$ | $f_I(s^\dagger)$ |
|---|---|---|---|---|---|---|---|---|---|---|---|---|---|
| $\alpha$ | / | $\uparrow$ | / | $\times$ | $\uparrow$ | $\uparrow$ | $\times$ | $\times$ | $\times$ | $\times$ | $\uparrow$ | $\uparrow$ | $\uparrow$ |
| $\beta$ | $\uparrow$ | $\downarrow$ | $\uparrow$ | $\uparrow$ | $\uparrow$ | / | $\uparrow$ | $\downarrow$ | / | $\downarrow$ | $\downarrow$ | $\uparrow$ | / |

# A. Omitted Proofs and Details from Section 3

This section provides proofs of the results in Section 3 and gives detailed derivations for Figure 1. Table 1 summarizes the notations used in this paper, Table 2 summarizes the monotonicity of key quantities with respect to abilities $\alpha, \beta$, and Figure 2 provides a flow chart for the workflow.

## A.1. Explicit forms of utility functions

Recall that

$$U_W(d,s) = f_W(s)d + g_W, \text{ and } U_I(d,s) = f_I(s)d + g_I.$$

We provide the explicit form of the coefficients below:

$$f_W(s) = (1 - p_a)\left((b_W + \ell_W)p_w - C_w\right)\phi(s) - C_v(s) - (b_W + \ell_W)(p_w - p_a) + C_w - C_a$$
$$g_W = (b_W + \ell_W)p_w - \ell_W - C_w$$
$$f_I(s) = (1 - p_a)\left((b_I + \ell_I)p_w - \xi C_w\right)\phi(s) - \xi C_v(s) - (b_I + \ell_I)(p_w - p_a) + \xi(C_w - C_a)$$
$$g_I = (b_I + \ell_I)p_w - \ell_I - \xi C_w.$$

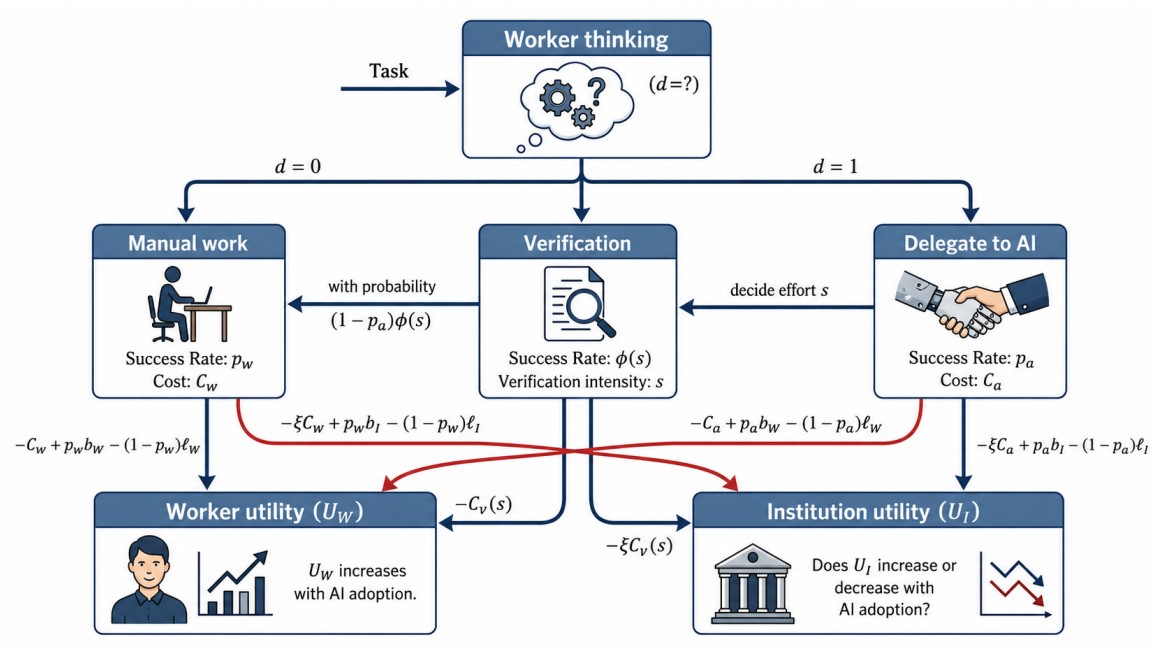

*Figure 2.* A flow chart for the workflow under AI assistance with worker action $(d, s) \in [0, 1]^2$.

Furthermore, note that $f_W(s) = \Phi(s; \alpha, \beta) + \Delta(\beta)$.

## A.2. An illustrative example for Assumption 3.3: monotonicity of detection

**Assumption A.1** (**Restatement of Assumption 3.3**). Assume that for any $\alpha, \beta \geq 0$, $\partial_\alpha \phi\big(s^\dagger(\alpha, \beta)\big) \geq 0$.

Consider the specific functional forms:

$$\phi(s; \alpha) := 1 - \frac{1}{1 + 10\alpha s}, \quad \text{and} \quad C_v(s) := 0.5s. \tag{8}$$

The constants 10 and 0.5 are chosen for convenience and can be replaced by other values, illustrating the generality of the assumption. Let $K_W(\beta) := (1 - p_a) [(b_W + \ell_W) p_w - C_w]$. The worker chooses effort $s \in [0, 1]$ to maximize the net benefit:

$$f_W(s) = K_W(\beta) \cdot \phi(s; \alpha) - C_v(s) + C_w - (b_W + \ell_W)(p_w - p_a) - C_a. \tag{9}$$

The first order condition with respect to $s$ is:

$$\frac{\partial f_W(s)}{\partial s} = K_W(\beta) \cdot \frac{10\alpha}{(1 + 10\alpha s)^2} - 0.5 = 0. \tag{10}$$

Solving for the interior root $s_0$, we obtain:

$$s_0(\alpha) = \sqrt{\frac{K_W(\beta)}{5\alpha}} - \frac{1}{10\alpha}. \tag{11}$$

It is straightforward to verify that $s_0(\alpha)$ is unimodal in $\alpha$ (increasing then decreasing), with a maximum value of $(s_0)_{\max} = K_W(\beta)/2$. The constrained optimal effort is $s^\dagger = \min(1, \max(0, s_0))$.

We now verify that the total derivative $\frac{d}{d\alpha}\phi(s^\dagger; \alpha)$ is strictly positive. Note that $\phi$ depends on $\alpha$ solely through the product $x := \alpha s$. Thus, $\phi(x) = 1 - (1 + 10x)^{-1}$, and $\frac{d\phi}{dx} = \frac{10}{(1+10x)^2} > 0$. It suffices to show that $\frac{d}{d\alpha}(\alpha s^\dagger) \geq 0$.

1. **Corner Solution ($s^\dagger = 0$):**
   In this regime, $s^\dagger = 0$, implying $\phi(0; \alpha) = 0$. The derivative is trivially non-negative.

2. **Corner Solution ($s^\dagger = 1$):**
   In this regime, $s^\dagger = 1$. The product becomes $x = \alpha$. The derivative is:

   $$\frac{d}{d\alpha}\phi(1;\alpha) = \frac{d\phi}{dx} \cdot \frac{dx}{d\alpha} = \frac{10}{(1+10\alpha)^2} \cdot 1 > 0. \tag{12}$$

3. **Interior Solution ($s^\dagger \in (0,1)$):**
   In this regime, $s^\dagger = s_0$. The product $x = \alpha s_0$ simplifies to:

   $$x(\alpha) = \alpha\left(\sqrt{\frac{K_W(\beta)}{5\alpha}} - \frac{1}{10\alpha}\right) = \sqrt{\frac{\alpha K_W(\beta)}{5}} - 0.1. \tag{13}$$

   Differentiating $x$ with respect to $\alpha$:

   $$\frac{dx}{d\alpha} = \frac{1}{2}\sqrt{\frac{K_W(\beta)}{5\alpha}} > 0. \tag{14}$$

   Consequently, by the Chain Rule:

   $$\partial_\alpha\phi(s^\dagger) = \frac{10}{(1+10x)^2} \cdot \frac{1}{2}\sqrt{\frac{K_W(\beta)}{5\alpha}} > 0. \tag{15}$$

Combining all cases, we conclude that $\frac{d}{d\alpha}\phi(s^\dagger) \geq 0$ for all $\alpha > 0$.

### A.3. Proof of Theorem 3.1: Worker optimal action

**Theorem A.2** (**Restatement of Theorem 3.1**). *Fix the task profile, AI characteristics, and baseline worker characteristics. Let $t \geq 0$ be a threshold such that $\Delta(t) = 0$. There exist continuous functions $\psi_0 : [t,\infty) \to \mathbb{R}_{\geq 0}$, monotonically increasing, and $\psi_1 : [0,t] \to \mathbb{R}_{\geq 0}$, monotonically decreasing, such that, for all $\beta$ in their respective domains,*

$$f_W\big(s^\dagger(\psi_0(\beta),\beta)\big) = 0 \qquad and \qquad \partial_s\Phi\big(0;\psi_1(\beta),\beta\big) = 0.$$

*Then the optimal action $(d^\star, s^\star)$ takes the following form:*

- *(**Manual work**) $(0,0)$ if $(\beta \geq t)$ and $(\alpha < \psi_0(\beta))$;*
- *(**Pure delegation**) $(1,0)$ if $(\beta < t)$ and $(\alpha < \psi_1(\beta))$;*
- *(**Verified delegation**) $(1, s^\dagger(\alpha,\beta))$ otherwise.*

*Proof.* We proceed by analyzing the properties of the threshold functions and then deriving the optimal actions for each regime.

**Monotonicity of $\psi_0(\beta)$.** Recall that $\psi_0(\beta)$ is defined implicitly by the condition $f_W(s^\dagger(\psi_0(\beta),\beta)) = 0$. Consider two values $\beta_1, \beta_2$ with $\beta_1 > \beta_2$. By Lemma 3.7, the function $f_W(s^\dagger(\alpha,\beta))$ is non-decreasing in $\alpha$ and non-increasing in $\beta$. Comparing the values at $(\psi_0(\beta_1),\beta_1)$ and $(\psi_0(\beta_1),\beta_2)$, we have:

$$0 = f_W\big(s^\dagger(\psi_0(\beta_1),\beta_1)\big) \leq f_W\big(s^\dagger(\psi_0(\beta_1),\beta_2)\big).$$

Since $f_W\big(s^\dagger(\psi_0(\beta_2),\beta_2)\big) = 0$, it follows that $f_W\big(s^\dagger(\psi_0(\beta_1),\beta_2)\big) \geq f_W\big(s^\dagger(\psi_0(\beta_2),\beta_2)\big)$. Because $f_W$ is non-decreasing in $\alpha$, this inequality implies $\psi_0(\beta_1) \geq \psi_0(\beta_2)$. Thus, $\psi_0(\beta)$ is monotonically increasing in $\beta$.

**Monotonicity of $\psi_1(\beta)$.** Recall that $\psi_1(\beta)$ is defined implicitly by $\partial_s\Phi(0;\psi_1(\beta),\beta) = 0$. Consider $\beta_1 > \beta_2$. By Lemma 3.8, the marginal benefit $\partial_s\Phi(0;\alpha,\beta)$ is non-decreasing in $\alpha$ and non-decreasing in $\beta$. Comparing the values at $(\psi_1(\beta_1),\beta_1)$ and $(\psi_1(\beta_1),\beta_2)$, we have:

$$0 = \partial_s\Phi(0;\psi_1(\beta_1),\beta_1) \geq \partial_s\Phi(0;\psi_1(\beta_1),\beta_2).$$

Since $\partial_s\Phi(0;\psi_1(\beta_2),\beta_2) = 0$, it follows that $\partial_s\Phi(0;\psi_1(\beta_1),\beta_2) \leq \partial_s\Phi(0;\psi_1(\beta_2),\beta_2)$. Because $\partial_s\Phi$ is non-decreasing in $\alpha$, this implies $\psi_1(\beta_1) \leq \psi_1(\beta_2)$. Thus, $\psi_1(\beta)$ is monotonically decreasing in $\beta$.

**Relationship between $f_W(0)$ and $t$.** The term $f_W(0)$ represents the net benefit of pure delegation relative to manual work. Note that $f_W(0) = C_w(\beta) - C_a - (b_W + \ell_W)(p_w(\beta) - p_a)$. Since the manual cost $C_w$ is decreasing in $\beta$, we have $\partial_\beta f_W(0) < 0$. The threshold $t$ is defined such that $\Delta(t) = f_W(0)|_{\beta=t} = 0$. Therefore:

- If $\beta \geq t$, then $f_W(0) \leq 0$.

- If $\beta < t$, then $f_W(0) > 0$.

**Optimal action regimes.** We now characterize the optimal strategy $(d^\star, s^\star)$ based on the sign of the optimal verification intensity $s^\dagger$ and the maximal value function $f_W(s^\dagger)$.

1. **Manual work** $(0,0)$**.** The worker chooses manual work if and only if the utility of delegation (even with optimal verification) does not exceed manual utility, i.e., $f_W(s^\dagger(\alpha, \beta)) \leq 0$. Since $f_W(s^\dagger)$ is maximized at $s^\dagger$ and bounded below by pure delegation $f_W(0)$, a necessary condition is $f_W(0) \leq 0$, which implies $\beta \geq t$. Given $\beta \geq t$, the condition $f_W(s^\dagger(\alpha, \beta)) \leq 0$ holds if and only if $\alpha \leq \psi_0(\beta)$ since $f_W$ is increasing in $\alpha$ and zero at $\psi_0$. Thus, the condition is: $\beta \geq t$ and $\alpha < \psi_0(\beta)$.

2. **Pure delegation** $(1,0)$**.** The worker chooses pure delegation if and only if verification is not worthwhile ($s^\dagger = 0$) and delegation is better than manual work ($f_W(0) > 0$). First, $f_W(0) > 0$ implies $\beta < t$. Second, $s^\dagger = 0$ corresponds to the corner solution where the marginal benefit of verification at zero is non-positive: $\partial_s \Phi(0; \alpha, \beta) \leq 0$. Since $\partial_s \Phi$ is increasing in $\alpha$ and zero at $\psi_1$, this requires $\alpha \leq \psi_1(\beta)$. Thus, the condition is: $\beta < t$ and $\alpha < \psi_1(\beta)$.

3. **Verified delegation** $(1, s^\dagger)$**.** The worker chooses verified delegation in all other cases, specifically when $s^\dagger > 0$ and $f_W(s^\dagger) > 0$. This occurs if:

   - $\beta < t$ and $\alpha > \psi_1(\beta)$ (Delegation is profitable, and verification is valuable).
   - $\beta \geq t$ and $\alpha > \psi_0(\beta)$ (Delegation is only profitable because verification is sufficiently valuable to overcome the manual baseline). Note that when $\beta \geq t$, we must have $\psi_0(\beta) \geq \psi_1(\beta)$ for a solution to exist, ensuring consistency.

This concludes the proof of the characterization. $\qquad\square$

### A.4. Proof of Theorem 3.5: Worker quality improvement v.s. no AI

**Theorem A.3** (**Restatement of Theorem 3.5**)**.** *Fix the task profile, AI characteristics, and baseline worker characteristics. Let $t$ and $\psi_0(\cdot)$ be as given in Theorem 3.1. Under Assumptions 3.2 and 3.3, there exists a continuous function $\psi : \mathbb{R}_{\geq 0} \to \mathbb{R}_{\geq 0}$ such that*

$$(f_I(s^\star) = 0) \wedge (d^\star = 1) \text{ for any pair } (\alpha, \beta) = (\psi(\beta), \beta).$$

*Moreover,*

- **(Improved)** $Q(W) > Q_0(W)$ *holds iff* $\alpha > \psi(\beta)$;
- **(Unchanged)** $Q(W) = Q_0(W)$ *holds iff* $\big((\beta \geq t) \wedge (\alpha < \psi_0(\beta))\big)$ *or* $\alpha = \psi(\beta)$;
- **(Degraded)** $Q(W) < Q_0(W)$ *holds otherwise.*

*Proof.* We analyze the relationship between the equilibrium quality $Q(W)$ and the baseline manual quality $Q_0(W)$. Recall that the difference is given by $Q(W) - Q_0(W) = f_I(s^\star) \cdot d^\star$. Thus, the deviation depends on the delegation decision $d^\star$ and the sign of the institution's net benefit $f_I(s^\star)$.

**Construction of $\psi(\beta)$.** We define the threshold function $\psi(\beta)$ implicitly by the condition that the institution breaks even under delegation. Specifically, for an ability pair $(\alpha, \beta) = (\psi(\beta), \beta)$, we satisfy $d^\star = 1$ and $f_I(s^\dagger(\psi(\beta), \beta)) = 0$. By Lemma 3.9, $f_I(s^\dagger(\alpha, \beta))$ is non-decreasing in $\alpha$, and the optimal delegation $d^\star$ is non-decreasing in $\alpha$. Given that $d^\star = 1$ at the threshold, it follows that for any $\alpha > \psi(\beta)$, we have $d^\star = 1$ and $f_I(s^\dagger) > 0$. Conversely, if $\alpha \leq \psi(\beta)$ and delegation occurs, $f_I(s^\dagger) \leq 0$.

**Improved ($Q(W) > Q_0(W)$).** This regime holds if and only if the worker delegates ($d^\star = 1$) and the institutional benefit is strictly positive ($f_I(s^\dagger) > 0$). Based on the monotonicity established above, $f_I(s^\dagger) > 0$ holds if and only if $\alpha > \psi(\beta)$. Since $\alpha > \psi(\beta)$ implies $d^\star = 1$ (delegation is optimal for the worker), the condition simplifies to:

$$\alpha > \psi(\beta).$$

**Unchanged ($Q(W) = Q_0(W)$).** This regime holds if either the worker chooses manual work ($d^\star = 0$) or delegates with zero net institutional gain ($d^\star = 1 \wedge f_I(s^\dagger) = 0$).

- From Theorem 3.1, manual work ($d^\star = 0$) is chosen if and only if $\beta \geq t$ and $\alpha < \psi_0(\beta)$.

- Verified delegation with zero gain occurs exactly at the threshold $\alpha = \psi(\beta)$.

Combining these disjoint cases yields the condition:

$$\big((\beta \geq t) \wedge (\alpha < \psi_0(\beta))\big) \vee (\alpha = \psi(\beta)).$$

**Degraded ($Q(W) < Q_0(W)$).** This regime holds if and only if the worker delegates ($d^\star = 1$) despite the institutional benefit being negative ($f_I(s^\dagger) < 0$). First, $f_I(s^\dagger) < 0$ implies $\alpha < \psi(\beta)$. Second, for the worker to delegate in this region, they must satisfy the delegation conditions from Theorem 3.1: either $(\beta < t)$ or $(\beta \geq t \wedge \alpha > \psi_0(\beta))$. Thus, degradation occurs if:

$$(\alpha < \psi(\beta)) \wedge \big[(\beta < t) \vee (\beta \geq t \wedge \alpha > \psi_0(\beta))\big].$$

$\square$

### A.5. Proof of Theorem 3.6: Worker upgraded v.s. downgraded

**Theorem A.4** (**Restatement of Theorem 3.6**)**.** *Fix the task profile, AI characteristics, and baseline worker characteristics. Fix a grading threshold $\tau \geq 0$. Let $t_\tau \geq 0$ be the solution to $Q_0(W; \beta) = \tau$. Under Assumptions 3.2 and 3.3, there exists a monotonically decreasing function $\psi_\tau : \mathbb{R}_{\geq 0} \to \mathbb{R}_{\geq 0}$ such that $Q(W) = \tau$ for the ability pair $(\alpha, \beta) = (\psi_\tau(\beta), \beta)$. Then*

- *(**Compliance gain**) $(Q(W) > \tau) \wedge (Q_0(W) < \tau)$ holds iff $\big(\beta < \min\{t, t_\tau\}\big) \wedge \big(\alpha > \psi_\tau(\beta)\big)$ or $\big(t \leq \beta < t_\tau\big) \wedge \big(\alpha > \max\{\psi_0(\beta), \psi_\tau(\beta)\}\big)$.*
- *(**Compliance loss**) $(Q(W) < \tau) \wedge (Q_0(W) > \tau)$ holds iff $\big(t_\tau < \beta \leq t\big) \wedge \big(\alpha < \psi_\tau(\beta)\big)$ or $\big(\beta > \max\{t, t_\tau\}\big) \wedge \big(\psi_0(\beta) < \alpha < \psi_\tau(\beta)\big)$.*

*Proof.* We characterize the regions where the worker's performance relative to the grading threshold $\tau$ changes due to AI availability.

**Threshold properties.** Let $t_\tau$ be the solution to $Q_0(W; \beta) = \tau$. Recall that $Q_0(W) = (b_I + \ell_I)p_W(\beta) - \ell_I - \xi C_w(\beta)$. Since the manual cost $C_w$ is decreasing in $\beta$, $Q_0(W)$ is strictly increasing in $\beta$. Therefore:

$$Q_0(W) > \tau \iff \beta > t_\tau \quad \text{and} \quad Q_0(W) < \tau \iff \beta < t_\tau.$$

Next, let $\psi_\tau(\beta)$ be the ability level $\alpha$ such that the delegated quality equals $\tau$, i.e., $Q(W)|_{d^\star = 1} = \tau$. By Lemma 3.10, $Q(W)$ under delegation is non-decreasing in $\alpha$. Thus:

$$Q(W)|_{d^\star = 1} > \tau \iff \alpha > \psi_\tau(\beta).$$

**Compliance Gain ($(Q(W) > \tau) \wedge (Q_0(W) < \tau)$).** This scenario implies the worker fails manually but succeeds with AI. First, $Q_0(W) < \tau$ necessitates $\beta < t_\tau$. Second, $Q(W) > \tau$ implies the worker must delegate ($d^\star = 1$), because manual work would yield $Q_0 < \tau$. Thus, we require $d^\star = 1$ and $\alpha > \psi_\tau(\beta)$. We analyze delegation in two sub-regions of $\beta$:

- If $\beta < t$: Delegation is always preferred. Combining with $\beta < t_\tau$, we have $\beta < \min\{t, t_\tau\}$. The only remaining condition is $\alpha > \psi_\tau(\beta)$.

- If $\beta \geq t$: Delegation occurs only if $\alpha > \psi_0(\beta)$. Combining with $\beta < t_\tau$, we have $t \leq \beta < t_\tau$. The condition becomes $\alpha > \max\{\psi_0(\beta), \psi_\tau(\beta)\}$.

Combining these yields:

$$\big(\beta < \min\{t, t_\tau\} \wedge \alpha > \psi_\tau(\beta)\big) \vee \big(t \leq \beta < t_\tau \wedge \alpha > \max\{\psi_0(\beta), \psi_\tau(\beta)\}\big).$$

**Compliance Loss** $((Q(W) < \tau) \wedge (Q_0(W) > \tau))$**.**  This scenario implies the worker succeeds manually but fails with AI. First, $Q_0(W) > \tau$ necessitates $\beta > t_\tau$. Second, $Q(W) < \tau$ implies the worker must delegate ($d^\star = 1$), because manual work would yield $Q_0 > \tau$. Thus, we require $d^\star = 1$ and $\alpha < \psi_\tau(\beta)$. We analyze delegation in two sub-regions of $\beta$:

- If $\beta \leq t$: Delegation is always preferred. Combining with $\beta > t_\tau$, we have $t_\tau < \beta \leq t$. The condition is simply $\alpha < \psi_\tau(\beta)$.

- If $\beta > t$: Delegation occurs only if $\alpha > \psi_0(\beta)$. Combining with $\beta > t_\tau$, we have $\beta > \max\{t, t_\tau\}$. The condition becomes $\psi_0(\beta) < \alpha < \psi_\tau(\beta)$.

Combining these yields:

$$\big(t_\tau < \beta \leq t \wedge \alpha < \psi_\tau(\beta)\big) \vee \big(\beta > \max\{t, t_\tau\} \wedge \psi_0(\beta) < \alpha < \psi_\tau(\beta)\big).$$

$\square$

### A.6. Proof of Lemma 3.7: effects of $\alpha, \beta$ on $f_W$

**Lemma A.5 (Restatement of Lemma 3.7).** $f_W\big(s^\dagger(\alpha, \beta)\big)$ *is non-decreasing in* $\alpha$ *and non-increasing in* $\beta$.

*Proof.* Recall the definition of the worker's net benefit from delegation:

$$f_W(s) = K_w(\beta)\,\phi(s; \alpha) - C_v(s) - C_a + C_w(\beta) - (b_W + \ell_W)(p_w - p_a),$$

where

$$K_w(\beta) := (1 - p_a)\big((b_W + \ell_W)p_w - C_w(\beta)\big).$$

Let $F_W(\alpha, \beta) := \max_{s \in [0,1]} f_W(s)$. Note that $K_w(\beta) \geq 0$ holds under the viability assumption.

**Monotonicity in $\alpha$.**  Fix $\beta$. For any fixed $s \in [0, 1]$, differentiating with respect to $\alpha$ yields:

$$\frac{\partial}{\partial \alpha} f_W(s) = K_w(\beta)\,\frac{\partial}{\partial \alpha}\phi(s; \alpha).$$

Since $K_w(\beta) \geq 0$ and the detection probability $\phi(s; \alpha)$ is non-decreasing in verification ability $\alpha$, we have $\frac{\partial}{\partial \alpha} f_W(s) \geq 0$. By the Envelope Theorem, since the objective function shifts upward pointwise, the maximum value $F_W(\alpha, \beta)$ must be non-decreasing in $\alpha$. Specifically, for $\alpha_2 > \alpha_1$:

$$F_W(\alpha_2, \beta) = f_W(s^\dagger_{\alpha_2} \mid \alpha_2) \geq f_W(s^\dagger_{\alpha_1} \mid \alpha_2) \geq f_W(s^\dagger_{\alpha_1} \mid \alpha_1) = F_W(\alpha_1, \beta).$$

**Monotonicity in $\beta$.**  Fix $\alpha$ and $s \in [0, 1]$. We differentiate $f_W(s)$ with respect to $\beta$. Note that $\beta$ affects $f_W$ primarily through the manual cost $C_w(\beta)$. Expanding the term involving $C_w$:

$$\text{Terms with } C_w(\beta) = -(1 - p_a)C_w(\beta)\phi(\alpha, s) + C_w(\beta) = C_w(\beta)\left[1 - (1 - p_a)\phi(s; \alpha)\right].$$

Differentiating with respect to $\beta$:

$$\frac{\partial}{\partial \beta} f_W(s) = \left[1 - (1 - p_a)\phi(s; \alpha)\right]\frac{\partial C_w(\beta)}{\partial \beta}.$$

First, observe the bracketed term. Since $\phi \in [0, 1]$ and $p_a \in [0, 1]$, we have $(1 - p_a)\phi \leq 1$, which implies $[1 - (1 - p_a)\phi] \geq 0$. Second, by assumption, manual cost decreases with skill $\beta$, so $\frac{\partial C_w}{\partial \beta} < 0$. Combining these, we obtain:

$$\frac{\partial}{\partial \beta} f_W(s) \leq 0.$$

Thus, $f_W(s)$ is pointwise non-increasing in $\beta$. It follows that the maximum value $F_W(\alpha, \beta)$ is non-increasing in $\beta$.  $\square$

**A.7. Proof of Lemma 3.8: effects of $\alpha, \beta$ on $\Phi$**

**Lemma A.6** (**Restatement of Lemma 3.8**). *$\partial_s \Phi(0; \alpha, \beta)$ is non-decreasing in $\alpha$ and non-decreasing in $\beta$.*

*Proof.* Recall the definition of the net verification benefit function:

$$\Phi(s; \alpha, \beta) := K_w(\beta)\phi(s; \alpha) - C_v(s),$$

where the coefficient $K_w(\beta)$ is defined as:

$$K_w(\beta) := (1 - p_a)\left((b_W + \ell_W)p_w - C_w(\beta)\right).$$

Recall that $K_w(\beta) \geq 0$ for all $\beta \geq 0$. Let $g(\alpha, \beta) := \partial_s \Phi(0; \alpha, \beta)$. Differentiating $\Phi$ with respect to $s$ and evaluating at $s = 0$, we obtain:

$$g(\alpha, \beta) = K_w(\beta) \cdot \partial_s \phi(0; \alpha) - C_v'(0).$$

**Monotonicity in $\alpha$.** Differentiating $g(\alpha, \beta)$ with respect to $\alpha$:

$$\frac{\partial}{\partial \alpha} g(\alpha, \beta) = K_w(\beta) \cdot \frac{\partial}{\partial \alpha}\left(\partial_s \phi(0; \alpha)\right).$$

Since $K_w(\beta) \geq 0$, the sign of the derivative depends on the term $\frac{\partial}{\partial \alpha}\partial_s \phi(0; \alpha)$. By definition of the derivative at zero and the property that $\phi(0) = 0$:

$$\partial_s \phi(0; \alpha) = \lim_{t \to 0^+} \frac{\phi(t; \alpha) - \phi(0; \alpha)}{t} = \lim_{t \to 0^+} \frac{\phi(t; \alpha)}{t}.$$

We know $\phi(t; \alpha)$ is non-decreasing in $\alpha$ for any fixed $t > 0$. Therefore, for $\alpha_2 > \alpha_1$, we have $\phi(t; \alpha_2) \geq \phi(t; \alpha_1)$, which implies:

$$\frac{\phi(t; \alpha_2)}{t} \geq \frac{\phi(t; \alpha_1)}{t}.$$

Since the inequality holds for all $t > 0$, it is preserved in the limit as $t \to 0$. Thus, $\partial_s \phi(0; \alpha)$ is non-decreasing in $\alpha$, implying $\frac{\partial}{\partial \alpha} g(\alpha, \beta) \geq 0$.

**Monotonicity in $\beta$.** Differentiating $g(\alpha, \beta)$ with respect to $\beta$:

$$\frac{\partial}{\partial \beta} g(\alpha, \beta) = \frac{\partial K_w(\beta)}{\partial \beta} \cdot \partial_s \phi(0; \alpha).$$

First, consider the derivative of $K_w(\beta)$. Assuming $p_w$ is constant or absorbs into the cost scaling, the dependence on $\beta$ comes from the manual cost $C_w(\beta)$:

$$\frac{\partial K_w(\beta)}{\partial \beta} = -(1 - p_a)\frac{\partial C_w(\beta)}{\partial \beta}.$$

Since the manual cost decreases with skill ($\frac{\partial C_w}{\partial \beta} < 0$) and $p_a \leq 1$, we have $\frac{\partial K_w}{\partial \beta} > 0$. Second, consider $\partial_s \phi(0)$. Since $\phi(t)$ is a probability increasing from $\phi(0) = 0$, we have $\phi(t) \geq 0$ for $t > 0$, implying:

$$\partial_s \phi(0) = \lim_{t \to 0} \frac{\phi(t)}{t} \geq 0.$$

Thus, $\frac{\partial}{\partial \beta} g(\alpha, \beta) \geq 0$. We conclude that $\partial_s \Phi(0; \alpha, \beta)$ is non-decreasing in $\beta$. $\qquad\square$

**A.8. Proof of Lemma 3.9: effect of $\alpha$ on $f_I(s^\star)$**

**Lemma A.7** (**Restatement of Lemma 3.8**). *$d^\star$ is non-decreasing in $\alpha$; and for ability pairs with $d^\star = 1$, $f_I(s^\star)$ is non-decreasing in $\alpha$.*

*Proof.* We address the two assertions sequentially.

**Monotonicity of $d^\star$.** Recall the optimal policy from Theorem 3.1: the worker delegates ($d^\star = 1$) if and only if the net benefit of delegation exceeds the manual baseline, i.e., $f_W(s^\dagger) > \max(0, f_W(0))$. From Lemma 3.7, the value function $f_W(s^\dagger(\alpha, \beta))$ is non-decreasing in $\alpha$. Consequently, if delegation is optimal for a given $\alpha_1$ (implying $f_W(s^\dagger(\alpha_1)) >$ threshold), it must also be optimal for any $\alpha_2 > \alpha_1$, as $f_W(s^\dagger(\alpha_2)) \geq f_W(s^\dagger(\alpha_1))$. Thus, $d^\star$ is non-decreasing in $\alpha$.

**Monotonicity of $f_I(s^\star)$ under delegation.** Assume $d^\star = 1$. We analyze the institutional utility $f_I(s^\star)$ by comparing it to the worker's utility $f_W(s^\star)$. Recall that the effective benefit coefficients for the institution and the worker are:

$$K_I(\beta) := (1 - p_a)\big((b_I + \ell_I)p_w - \xi C_w(\beta)\big), \quad K_W(\beta) := (1 - p_a)\big((b_W + \ell_W)p_w - C_w(\beta)\big).$$

Using these coefficients, the utility functions under delegation ($d = 1$) can be written as:

$$f_W(s) = K_W(\beta)\phi(s; \alpha) - C_v(s) + \Delta_W,$$
$$f_I(s) = K_I(\beta)\phi(s; \alpha) - \xi C_v(s) + \Delta_I,$$

where $\Delta_W$ and $\Delta_I$ are terms independent of $s$ and $\alpha$. Observe that we can express $f_I(s)$ as a linear combination of $f_W(s)$ and the detection probability $\phi(s)$:

$$f_I(s) = \xi f_W(s) + (K_I(\beta) - \xi K_W(\beta))\phi(s; \alpha) + \text{Constant}.$$

We now differentiate the equilibrium value $f_I(s^\star)$ with respect to $\alpha$.

1. **Case 1: Pure delegation ($s^\star = 0$).** In this case, $s^\star$ is constant locally. Since $\phi(0; \alpha) = 0$, $f_I(0)$ is constant with respect to $\alpha$. Thus, $\frac{d}{d\alpha}f_I(s^\star) = 0$.

2. **Case 2: Verified delegation ($s^\star > 0$).** In this case, $s^\star = s^\dagger(\alpha, \beta)$. Differentiating the linked expression with respect to $\alpha$:

$$\frac{d}{d\alpha}f_I(s^\dagger) = \xi \frac{d}{d\alpha}f_W(s^\dagger) + (K_I(\beta) - \xi K_W(\beta))\frac{d}{d\alpha}\phi(s^\dagger; \alpha).$$

   We analyze the signs of these terms:

   - $\frac{d}{d\alpha}f_W(s^\dagger) \geq 0$ by Lemma 3.7.
   - By Assumption 3.2, $(b_I + \ell_I) \geq \xi(b_W + \ell_W)$, which implies $K_I(\beta) \geq \xi K_W(\beta)$. Since $K_W > 0$, the coefficient $(K_I - \xi K_W)$ is non-negative.
   - The term $\frac{d}{d\alpha}\phi(s^\dagger; \alpha)$ represents the total change in equilibrium detection probability. By Assumption 3.3, the equilibrium detection probability is non-decreasing in $\alpha$.

   Combining these non-negative terms, we conclude that $\frac{d}{d\alpha}f_I(s^\star) \geq 0$.

$\square$

## A.9. Proof of Lemma 3.10: effects of $\alpha, \beta$ on $Q(W)$

**Lemma A.8** (**Restatement of Lemma 3.10**). *For ability pairs with $d^\star = 1$, worker quality $Q(W)$ is non-decreasing in $\alpha$ and non-decreasing in $\beta$.*

*Proof.* Let $G(\alpha, \beta) := Q(W)|_{d^\star=1}$. Using the effective benefit coefficients defined in the proof of Lemma 3.9, we write the institution's quality and the worker's objective function under delegation as:

$$G(\alpha, \beta) = K_I(\beta)\phi(s^\dagger; \alpha) - \xi C_v(s^\dagger) + \Gamma_I,$$
$$f_W(s^\dagger) = K_W(\beta)\phi(s^\dagger; \alpha) - C_v(s^\dagger) + \Gamma_W,$$

where $\Gamma_I, \Gamma_W$ are constants with respect to $s$ and $\alpha$. Recall that $K_I(\beta) \geq \xi K_W(\beta) > 0$, and $\frac{dK_I}{d\beta} > 0$, $\frac{dK_W}{d\beta} > 0$ (due to decreasing manual costs).

**Monotonicity of $s^\dagger$ in $\beta$.** We first establish that the optimal effort $s^\dagger$ is non-decreasing in $\beta$. Consider the interior first order condition for the worker:

$$g(s, \beta) := \frac{\partial f_W}{\partial s} = K_W(\beta)\phi_s(s; \alpha) - C_v'(s) = 0.$$

By the implicit function theorem, the sensitivity of the interior solution $s_0$ is:

$$\frac{\partial s_0}{\partial \beta} = -\frac{\partial g/\partial \beta}{\partial g/\partial s}.$$

The denominator $\frac{\partial g}{\partial s} = \frac{\partial^2 f_W}{\partial s^2} < 0$ by the second-order condition for a maximum. The numerator is $\frac{\partial g}{\partial \beta} = K_W'(\beta)\partial_s\phi(s)$. Since $K_W'(\beta) > 0$ and $\partial_s\phi(s) > 0$, the numerator is positive. Therefore, $\frac{\partial s_0}{\partial \beta} > 0$. Since $s^\dagger = \min(1, \max(0, s_0))$, the monotonicity holds for the constrained solution as well.

**Monotonicity of $Q(W)$ in $\beta$.** Differentiating $G(\alpha, \beta)$ with respect to $\beta$:

$$\frac{\partial G}{\partial \beta} = K_I'(\beta)\phi(s^\dagger) + \frac{\partial s^\dagger}{\partial \beta}\left[K_I(\beta)\partial_s\phi(s^\dagger) - \xi C_v'(s^\dagger)\right].$$

From the worker's FOC, we know that for an interior solution, $C_v'(s^\dagger) = K_W(\beta)\partial_s\phi(s^\dagger)$. If $s^\dagger = 0$, the derivative term vanishes or is positive; if $s^\dagger = 1$, $\frac{\partial s^\dagger}{\partial \beta} = 0$. Focusing on the interior case, we substitute $C_v'$:

$$\frac{\partial G}{\partial \beta} = K_I'(\beta)\phi(s^\dagger) + \frac{\partial s^\dagger}{\partial \beta}\phi_s(s^\dagger)\left[K_I(\beta) - \xi K_W(\beta)\right].$$

All terms are non-negative:

- $K_I'(\beta) \geq 0$ (decreasing manual cost increases net institutional benefit).

- $\frac{\partial s^\dagger}{\partial \beta} \geq 0$ (from above).

- $K_I(\beta) \geq \xi K_W(\beta)$ (Assumption 3.2).

Thus, $\frac{\partial G}{\partial \beta} \geq 0$.

**Monotonicity of $Q(W)$ in $\alpha$.** Differentiating $G(\alpha, \beta)$ with respect to $\alpha$:

$$\frac{\partial G}{\partial \alpha} = K_I(\beta)\left[\partial_\alpha\phi(s^\dagger) + \partial_s\phi(s^\dagger)\frac{\partial s^\dagger}{\partial \alpha}\right] - \xi C_v'(s^\dagger)\frac{\partial s^\dagger}{\partial \alpha}.$$

Rearranging terms:

$$\frac{\partial G}{\partial \alpha} = K_I(\beta)\partial_\alpha\phi(s^\dagger) + \frac{\partial s^\dagger}{\partial \alpha}\left[K_I(\beta)\partial_s\phi(s^\dagger) - \xi C_v'(s^\dagger)\right].$$

Again, using the worker's first-order condition $C_v' \leq K_W\partial_s\phi$, which implies that $K_I\partial_s\phi - \xi C_v' \geq K_I\phi_s - \xi K_W\phi_s$:

$$\frac{\partial G}{\partial \alpha} \geq K_I(\beta)\partial_\alpha\phi(s^\dagger) + \frac{\partial s^\dagger}{\partial \alpha}\partial_s\phi(s^\dagger)\left[K_I(\beta) - \xi K_W(\beta)\right].$$

Since $\partial_\alpha\phi \geq 0$, $\frac{\partial s^\dagger}{\partial \alpha} \geq 0$ (Lemma 3.9), and $K_I \geq \xi K_W$, the total derivative is non-negative. □

### A.10. Details of the illustrative example in Figure 1

We first illustrate the choice of parameters and functions in Figure 1. Consider a worker with a linear verification cost $C_v(s) = s$, an execution cost $C_w(\beta) = 5(1 - \beta)$ for efficiency $\beta \in [0, 1]$ (where the factor 5 reflects that execution is substantially more costly than verification), and an inverse-linear error-detection function $\phi(s; \alpha) = 1 - \frac{1}{1+2\alpha s}$. To match the scale of $\beta$, we restrict to $\alpha \in [0, 1]$, where the constant 2 serves as a scaling parameter for this normalization. Below, we select a representative parameter setting for analysis.

*Task profile.* We set $b_W = 8$, $\ell_W = 6$, $b_I = 14$, $\ell_I = 12$, $\xi = 0.3$, and $\tau = 6.4$, which satisfies Assumption 3.2.

*AI characteristics.* We set $p_a = 0.65$ (moderate reliability) and $C_a = 0$ (negligible execution cost).

*Worker characteristics.* We set $p_w = 0.75$, so that the worker outperforms the AI in task success probability.

**Characterization of key quantities.** We next compute the key quantities that govern the regime characterization in our theoretical results.

*To Theorem 3.1.* To compute the threshold $t$, we first obtain

$$\Delta(\beta) = 5(1 - \beta) - C_a - (b_W + \ell_W)(p_w - p_a).$$

Under the above parameter choices, this yields

$$t = 1 - \frac{C_a + (b_W + \ell_W)(p_w - p_a)}{5} = 0.72. \tag{16}$$

We also derive closed-form expressions for the two regime boundaries (on their respective domains):

$$\psi_0(\beta) = \frac{20}{\left(\sqrt{77 + 70\beta} - \sqrt{200\beta - 144}\right)^2}, \qquad \psi_1(\beta) = \frac{20}{77 + 70\beta}.$$

Finally, within the verified-delegation regime, the optimal action takes the form $(d^\star, s^\star) = (1, s^\dagger(\alpha, \beta))$ with

$$s^\dagger(\alpha, \beta) = \sqrt{\frac{1.925 + 1.75\beta}{\alpha}} - \frac{1}{\alpha}.$$

*To Theorem 3.5.* The key is to characterize the separatrix $\psi$. To this end, we first define a function $\psi' : [0, 1] \to [0, 1]$ (monotonically decreasing in $\beta$) such that

$$f_I\big(s^\dagger(\alpha, \beta)\big) = 0 \text{ for any ability pair } (\alpha, \beta) = (\psi'(\beta), \beta).$$

Extending $\psi_0(\beta) = 0$ for $\beta < t$, Theorem 3.5 implies that $\psi(\beta) = \max\{\psi_0(\beta), \psi'(\beta)\}$. Finally, the closed-form condition of $f_I\big(s^\dagger(\alpha, \beta)\big) = 0$ is:

$$5.2 - 0.975\beta = \frac{6.3 + 0.525\beta}{10\sqrt{\alpha(0.077 + 0.07\beta)} - 1} + \frac{15\sqrt{\alpha(0.077 + 0.07\beta)} - 3}{10\alpha}$$

*To Theorem 3.6.* To compute the threshold $t_\tau$, we first obtain

$$Q_0(W) = g_I = (b_I + \ell_I)p_w - \ell_I - 5\xi(1 - \beta).$$

Solving $Q_0(W) = \tau$ yields the threshold $t_\tau = \frac{4}{15}$. under our parameter choices. We also derive a closed-form condition for $Q(W) = \tau$, which defines the separatrix $\psi_\tau$:

$$\frac{(252 + 21\beta)(2 - \sqrt{7\alpha(1.1 + \beta)})}{40(\sqrt{7\alpha(1.1 + \beta)} - 1)} - \frac{3\sqrt{1.75(1.1 + \beta)}}{10\sqrt{\alpha}} + \frac{0.3}{\alpha} + 4.9 = \tau.$$

**Interpretation of the parameterization.** The parameters in Figure 1 are chosen as an illustrative setting that makes the model's threshold and quality-regime phenomena visible. They encode a structurally asymmetric environment: the institution has larger stakes in success and failure, internalizes only a discounted share of the worker's private effort cost, and faces an AI system that is cheaper but less accurate than manual execution. Thus, delegation can be privately attractive even when it is institutionally harmful.

The cutoff is set to $\tau = 6.4$. Since the pre-AI baseline quality is $Q_0(W; \beta) = 6 + 1.5\beta$, the baseline qualification cutoff is $\beta = 4/15$. Hence, the baseline excludes only low-efficiency workers in $\beta \in [0, 1]$, allowing the figure to focus on how AI-induced delegation and verification reshape qualification outcomes.

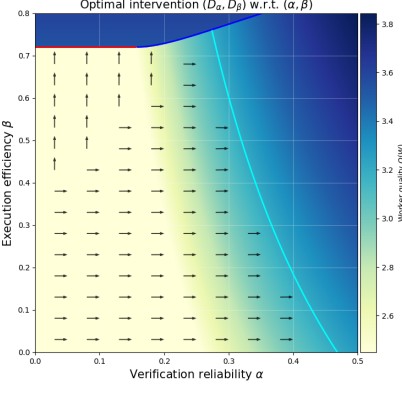

*(a)* Optimal intervention $(D_\alpha, D_\beta)$

*Figure 3.* Plots illustrating the effects of the worker-side intervention, as functions of abilities $(\alpha, \beta)$, under the default parameter setting $(b_W, \ell_W, b_I, \ell_I, \xi, \tau, p_a, C_a, p_w) = (8, 6, 14, 12, 0.3, 6.4, 0.65, 0, 0.75)$, and the functional choices $C_v(s) = s$, $C_w(\beta) = 5(1 - \beta)$, $\phi(s; \alpha) = 1 - \frac{1}{1+2\alpha s}$, and $h_1(x) = h_2(x) = x$. We restrict the domain to $[0, 0.5] \times [0, 0.8]$ for clarity.

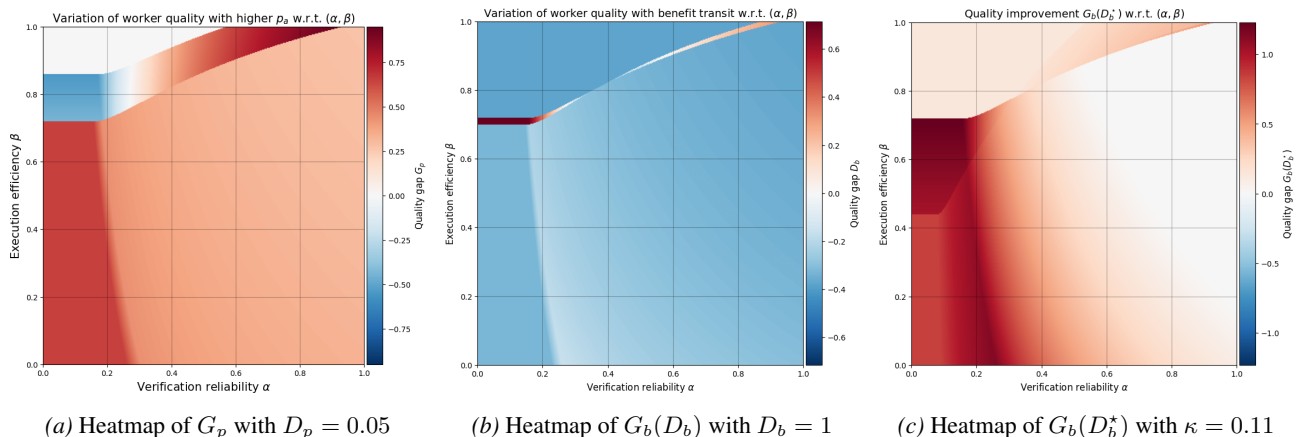

*(a)* Heatmap of $G_p$ with $D_p = 0.05$  *(b)* Heatmap of $G_b(D_b)$ with $D_b = 1$  *(c)* Heatmap of $G_b(D_b^\star)$ with $\kappa = 0.11$

*Figure 4.* Plots illustrating the effects of the institutional-side interventions, as functions of abilities $(\alpha, \beta)$, under the default parameter setting $(b_W, \ell_W, b_I, \ell_I, \xi, \tau, p_a, C_a, p_w) = (8, 6, 14, 12, 0.3, 6.4, 0.65, 0, 0.75)$, and the functional choices $C_v(s) = s$, $C_w(\beta) = 5(1 - \beta)$, $\phi(s; \alpha) = 1 - \frac{1}{1+2\alpha s}$. In Figures 4a, 4b, and 4c , we analyze how worker quality varies under a more capable AI, under a larger marginal gain, and under the institution's dynamic strategy, respectively.

## B. Interventions to Improve Worker Quality

The previous analysis shows that AI-assisted work can generate quality loss when workers rationally delegate without sufficient verification. This section studies how such losses can be mitigated through interventions. We distinguish two broad classes of interventions. The first class is worker-side: it improves the worker's underlying abilities, such as verification reliability $\alpha$ or execution efficiency $\beta$. The second class is institutional-side: it changes the environment in which the worker chooses a workflow, such as AI capability, payoff incentives, or direct rewards for verification.

These two classes correspond to different policy questions. Worker-side interventions ask how to improve a given worker's ability profile at minimum cost so that the resulting quality reaches a target level. Institutional-side interventions ask how changes in AI systems or incentives reshape the worker's optimal delegation and verification behavior, and whether these changes improve institutional quality. A central message is that intervention effects are heterogeneous: improving a primitive such as AI capability or worker incentives need not uniformly improve quality, because the intervention may also induce workers to switch workflow regimes.

## B.1. Worker-side interventions

It follows from Lemma 3.10 that increasing either verification reliability $\alpha$ or execution efficiency $\beta$ can improve worker quality $Q(W)$, and thus both can serve as worker-side interventions. To determine which intervention is more cost-effective, we formulate a constrained optimization problem.

We consider two interventions: increasing $\alpha$ by $D_\alpha \geq 0$ and increasing $\beta$ by $D_\beta \geq 0$. Let $Q(D_\alpha, D_\beta)$ denote the resulting worker quality under abilities $(\alpha + D_\alpha, \beta + D_\beta)$. The goal is to ensure $Q(D_\alpha, D_\beta) \geq \tau$ while minimizing intervention cost.

Let $h_1, h_2 : \mathbb{R}_{\geq 0} \to \mathbb{R}_{\geq 0}$ be monotone functions that model the costs of increasing $\alpha$ and $\beta$, respectively. A simple specification is linear, $h_i(x) = c_i x$ for $i \in \{1, 2\}$, which is justified as a first-order approximation when $x$ is small. More generally, one may take $h_i(x) = c_i x^\rho$ for $\rho > 1$, capturing increasing marginal costs of continued upskilling.

**Optimization problem.** We formalize the intervention design as:

$$\min_{D_\alpha, D_\beta \geq 0} h_1(D_\alpha) + h_2(D_\beta) \quad \text{s.t.} \quad Q(D_\alpha, D_\beta) \geq \tau. \tag{17}$$

**Analysis.** We use the same setting as in Figure 1 and set $h_1(x) = h_2(x) = x$. Figure 3 plots the optimal intervention ways for workers with various $(\alpha, \beta)$. Observe that improving $\alpha$ is usually more cost-efficient, validating the central role of verification reliability in the AI age.

## B.2. Institutional-side interventions

On the institutional-side, we propose three potential interventions: deploying more capable AI systems, reshaping worker incentives, and directly rewarding verification. Below we illustrate these three interventions.

**Deploying more capable AI systems.** This intervention can be implemented by training a task-specific agent and corresponds to increasing the AI success probability $p_a$. Under this intervention, both the worker's optimal action and the resulting quality may change. Let $D_p \in [0, 1 - p_a]$ denote the increase in $p_a$, and let $Q(D_p)$ denote the worker quality under AI capability $p_a + D_p$. We define the *quality variation value* as

$$G_p(D_p) := Q(D_p) - Q(0).$$

Figure 4a plots a heatmap of $G_p$ with $D_p = 0.05$. We observe that $G_p$ can be positive in some regions of $(\alpha, \beta)$ but negative in others, indicating that improving AI capability may either increase or decrease worker quality depending on worker abilities. Thus, deploying more capable AI systems can have a non-monotonic effect on worker quality. Specifically, workers with high execution efficiency $\beta$ but low verification reliability $\alpha$ may switch from manual work to pure delegation, which can reduce quality.

**Incentive reshaping and equilibrium design.** Another institutional lever is to reshape the worker's payoff so that the worker internalizes a larger share of the task-success benefit. We consider two related formulations.

*Fixed incentive reshaping.* We first consider a fixed incentive transfer. Let $D_b \geq 0$ denote an exogenously specified transfer from the institution to the worker: the worker's success benefit becomes $b_W + D_b$, while the institution's marginal success benefit becomes $b_I - D_b$. Let $Q(D_b)$ denote the worker quality induced by the worker's optimal action under the reshaped incentive $b_W + D_b$, and define

$$G_b(D_b) := Q(D_b) - Q(0).$$

Figure 4b plots $G_b(D_b)$ with $D_b = 1$. The figure shows that a fixed transfer has heterogeneous effects across the ability space. In some regions, increasing the worker's private success benefit improves institutional quality by making unchecked delegation less attractive and strengthening incentives to verify AI outputs. In particular, workers with relatively high execution efficiency but low verification reliability may move away from institutionally harmful pure delegation, thereby increasing quality. However, the same fixed transfer can also reduce institutional quality in other regions, since it lowers the institution's own marginal success benefit through $b_I - D_b$ and may induce behavioral changes whose institutional value is limited. Thus, fixed incentive reshaping is not uniformly beneficial; its effect depends on worker abilities and on the induced workflow response.

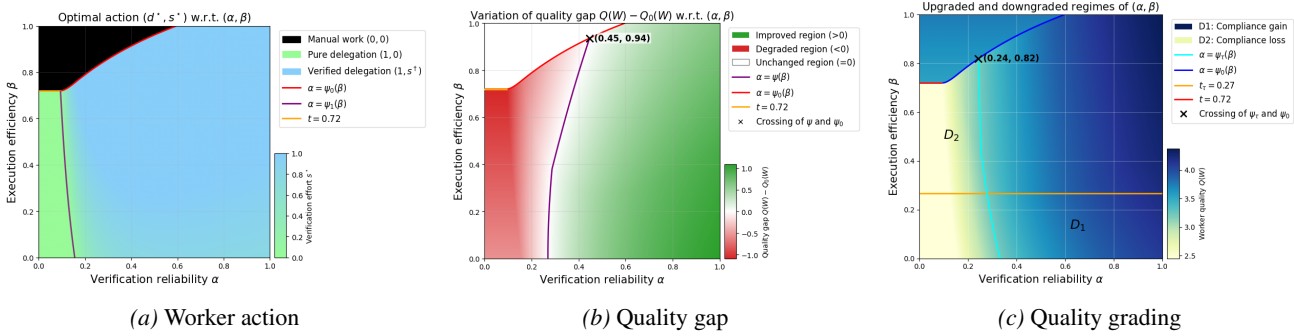

*(a)* Worker action  *(b)* Quality gap  *(c)* Quality grading

*Figure 5.* Plots illustrating the worker action and quality regimes under a linear verification reward $\gamma s$ (with $\gamma = 0.2$), as functions of abilities of $(\alpha, \beta)$, under the default parameter setting $(b_W, \ell_W, b_I, \ell_I, \xi, \tau, p_a, C_a, p_w) = (8, 6, 14, 12, 0.3, 6.4, 0.65, 0, 0.75)$, and the functional choices $C_v(s) = s$, $C_w(\beta) = 5(1 - \beta)$, and $\phi(s; \alpha) = 1 - \frac{1}{1 + 2\alpha s}$.

*Equilibrium incentive design.* The preceding analysis treats $D_b$ as an exogenously fixed policy. We next consider a distinct, richer design problem in which the transfer is chosen strategically by the institution. For a worker type $(\alpha, \beta)$, the institution first selects a transfer $D_b$, anticipating that the worker will subsequently re-optimize delegation and verification behavior under the reshaped benefit $b_W + D_b$. In this extension, we also introduce an additional alignment return

$$C(D_b) := \kappa D_b^2,$$

which captures the positive institutional value generated by higher worker motivation and stronger incentive alignment. The institution therefore solves

$$D_b^\star(\alpha, \beta) \in \arg\max_{D_b \geq 0} \big\{ Q(D_b) + C(D_b) \big\},$$

The worker then chooses its utility-maximizing action under $b_W + D_b^\star$. Then we define

$$G_b(D_b^\star) = Q(D_b^\star) + C(D_b^\star) - Q(0).$$

Under this augmented objective, the transfer becomes an endogenous institutional design variable rather than a fixed policy parameter. Figure 4c plots $G_b(D_b^\star)$ with $\kappa = 0.11$. The figure illustrates the resulting equilibrium gain under this design objective. The gain is most visible for workers originally located in the manual-work or pure-delegation regimes. For workers in the manual-work regime, the transfer can raise the private return to successful AI-assisted work and make verified delegation worthwhile. For workers in the pure-delegation regime, it can increase the private value of careful verification and thereby mitigate rational over-delegation. Thus, the equilibrium-design extension illustrates how strategic institutions may use incentive transfers as an alignment mechanism, inducing workflow choices that better preserve institutional quality in AI-assisted work.

**Rewarding observable verification.** If the institution can observe verification effort or workflow traces, it may reward the worker directly for verification rather than only for final outcomes. Let $R(s) \geq 0$ denote a process-based verification reward, paid only when the worker delegates and verifies the AI output. The worker's utility becomes

$$U_W^R(d, s) = U_W(d, s) + dR(s),$$

while the institutional utility becomes

$$U_I^R(d, s) = U_I(d, s) - dR(s).$$

For example, taking $R(s) = \gamma s$ directly increases the marginal private return to verification. This intervention therefore makes pure delegation less attractive relative to verified delegation, especially for workers whose verification reliability is moderate but whose baseline incentives to verify are weak.

Figure 5 illustrates this effect with a linear verification reward $R(s) = \gamma s$ and $\gamma = 0.2$. Compared to Figure 1, the verified-delegation regime in Figure 5a expands, indicating that the verification reward raises the worker's private incentive to scrutinize AI outputs. Correspondingly, Figure 5b shows that the quality-degradation region shrinks, suggesting that observable verification rewards can mitigate harmful over-delegation. Figure 5c further shows that this intervention can reduce compliance loss for some workers, although its effect remains heterogeneous across worker abilities.

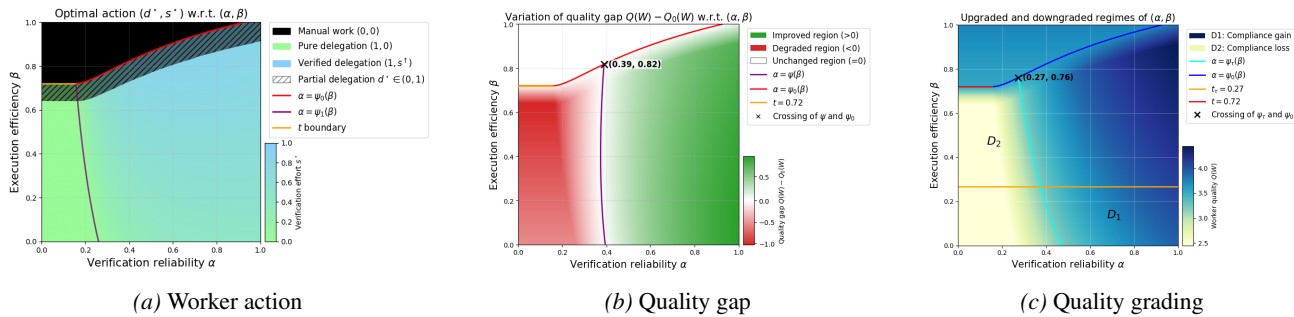

*Figure 6.* Plots illustrating the worker action and quality regimes under an additional quadratic delegation cost $0.1d^2$, as functions of abilities of $(\alpha, \beta)$, under the default parameter setting $(b_W, \ell_W, b_I, \ell_I, \xi, \tau, p_a, C_a, p_w) = (8, 6, 14, 12, 0.3, 6.4, 0.65, 0, 0.75)$, and the functional choices $C_v(s) = s$, $C_w(\beta) = 5(1 - \beta)$, and $\phi(s; \alpha) = 1 - \frac{1}{1+2\alpha s}$.

## C. Model Extensions

The baseline model isolates the delegation–verification mechanism under a parsimonious set of assumptions. In this section, we examine whether the main qualitative conclusions are robust when these assumptions are relaxed. Across the extensions, we focus on three conclusions from the baseline analysis: the phase-transition structure of worker actions, the coexistence of compliance gain and compliance loss, and the possibility of rational over-delegation.

We organize the extensions according to which component of the baseline model is perturbed. First, we relax the worker's action space. Section C.1 allows the worker to choose an interior level of delegation through a nonlinear delegation cost, while Section C.2 introduces time or cognitive resource constraints that restrict feasible workflows. Second, we relax the information structure: Section C.3 allows the worker to make delegation decisions under miscalibrated beliefs about AI capability. Third, we relax the task environment by allowing task difficulty to vary across instances in Section C.4.

We then consider richer delegated workflows. Section C.5 allows detected AI errors to be corrected through partial rather than full re-execution, and Section C.6 allows the worker to interact with the AI over multiple verification rounds. Finally, Sections C.7 and C.8 relax two modeling abstractions: binary outcomes and regularity assumptions on the verification technology.

The results below show that the main conclusions of the baseline model do not depend on any single simplifying assumption. The extensions may smooth regime boundaries, change the size of quality-gain or quality-loss regions, or alter the worker's optimal verification intensity, but they preserve the core mechanism: AI assistance improves quality only when delegation is paired with sufficiently reliable verification; otherwise, rational worker behavior can still generate institutional quality loss.

### C.1. Partial delegation via nonlinear delegation cost

The baseline specification makes worker utility affine in the delegation rate, $U_W(d, s) = f_W(s)d + g_W$, and therefore induces corner solutions in $d$. To examine whether the regime structure depends on this binary reliance property, we perturb the worker decision layer by adding a convex private cost of delegation intensity. For $\delta > 0$, let

$$U_W^\delta(d, s) := f_W(s)d + g_W - \delta d^2, \qquad (d, s) \in [0, 1]^2.$$

The additional term captures nonlinear cognitive or workflow-adjustment frictions from integrating AI into only part of task execution. The task-success function and institutional utility are unchanged. Let

$$(d_\delta^\star, s_\delta^\star) \in \arg \max_{(d,s) \in [0,1]^2} U_W^\delta(d, s), \qquad Q_\delta(W) := U_I(d_\delta^\star, s_\delta^\star).$$

For any fixed verification effort $s$, the perturbed objective is strictly concave in $d$, so the optimal delegation rate is

$$d_\delta^\star(s) = \Pi_{[0,1]}\left(\frac{f_W(s)}{2\delta}\right),$$

where $\Pi_{[0,1]}$ denotes projection onto $[0, 1]$. Moreover, the value after optimizing over $d$ is monotone in $f_W(s)$, so the

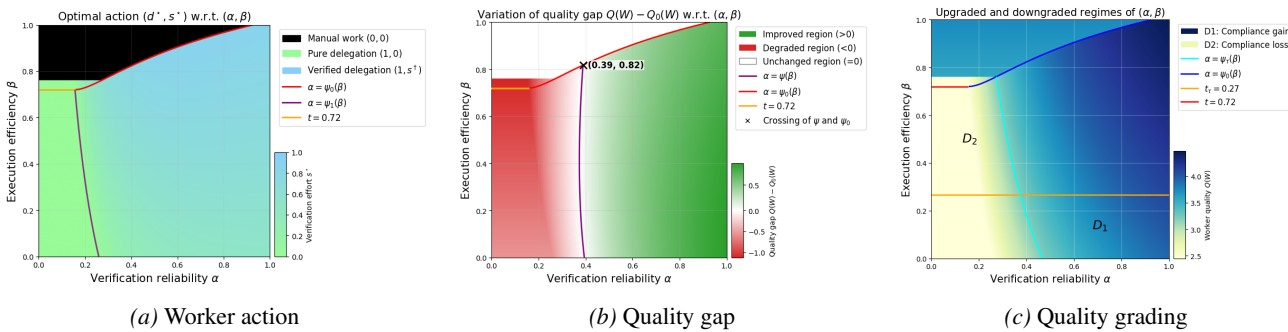

*(a)* Worker action        *(b)* Quality gap        *(c)* Quality grading

*Figure 7.* Plots illustrating the worker action and quality regimes under a cost restriction of $\text{cost}(d, s) = (1 - d)C_w + dC_v(s) \leq \tau_c$ (with $\tau_c = 0.6$), as functions of abilities of $(\alpha, \beta)$, under the default parameter setting $(b_W, \ell_W, b_I, \ell_I, \xi, \tau, p_a, C_a, p_w) = (8, 6, 14, 12, 0.3, 6.4, 0.65, 0, 0.75)$, and the functional choices $C_v(s) = s$, $C_w(\beta) = 5(1 - \beta)$, and $\phi(s; \alpha) = 1 - \frac{1}{1 + 2\alpha s}$.

relevant verification choice is still governed by

$$s^\dagger(\alpha, \beta) \in \arg \max_{s \in [0,1]} f_W(s),$$

whenever delegation is used. Hence the worker chooses no delegation when $f_W(s^\dagger) \leq 0$, partial delegation when $0 < f_W(s^\dagger) < 2\delta$, and full delegation when $f_W(s^\dagger) \geq 2\delta$. The perturbation does not introduce a new verification object, but inserts an interior feasible reliance layer between manual work and full AI reliance.

**Analysis.** Figure 6 visualizes the perturbed model with $\delta = 0.1$. In Figure 6a, the black-striped region marks the new partial-delegation layer, where the worker has positive delegation surplus but not enough to choose $d = 1$ once the quadratic reliance friction is included. Outside this layer, the action partition remains close to the baseline: low-verification workers may still purely delegate, sufficiently reliable workers enter verified delegation, and workers with nonpositive delegation surplus remain in manual work. Figures 6b and 6c show that the quality-improvement/degradation and compliance-gain/loss regimes also persist. Since

$$Q_\delta(W) - Q_0(W) = f_I(s_\delta^\star)d_\delta^\star,$$

partial delegation attenuates the magnitude of institutional gains or losses when $0 < d_\delta^\star < 1$, but does not remove the sign mismatch between worker and institutional objectives. Overall, this perturbation smooths the baseline corner transition in $d$ while preserving the core qualitative mechanism: AI can still improve workers with sufficiently reliable verification and degrade quality for workers who rationally delegate despite weak institutional value.

Together, these patterns show that the baseline regime structure is not an artifact of binary delegation. Allowing interior reliance smooths the transition between manual work and full delegation, but it does not eliminate the underlying misalignment between the worker's private delegation surplus and the institution's quality objective.

### C.2. Time and cognitive resource constraints

We next relax the assumption that all workflows are feasible. In practice, workers may face time, attention, or cognitive-resource limits that make manual execution or careful verification difficult. We model this by imposing a resource constraint on the worker's feasible reliance strategies.

Let $\tau_c \geq 0$ denote the worker's available resource capacity. Instead of changing the success function or the payoff primitives, we restrict the feasible set by

$$\mathcal{A}_{\tau_c} := \left\{ (d, s) \in [0, 1]^2 : (1 - d)C_w + dC_v(s) \leq \tau_c \right\}.$$

Here the resource budget captures the ex ante time or attention required to choose and implement a workflow, rather than the realized ex post correction cost after an AI error is detected. The worker's decision is therefore perturbed to

$$(d_{\tau_c}^\star, s_{\tau_c}^\star) \in \arg \max_{(d,s) \in \mathcal{A}_{\tau_c}} U_W(d, s).$$

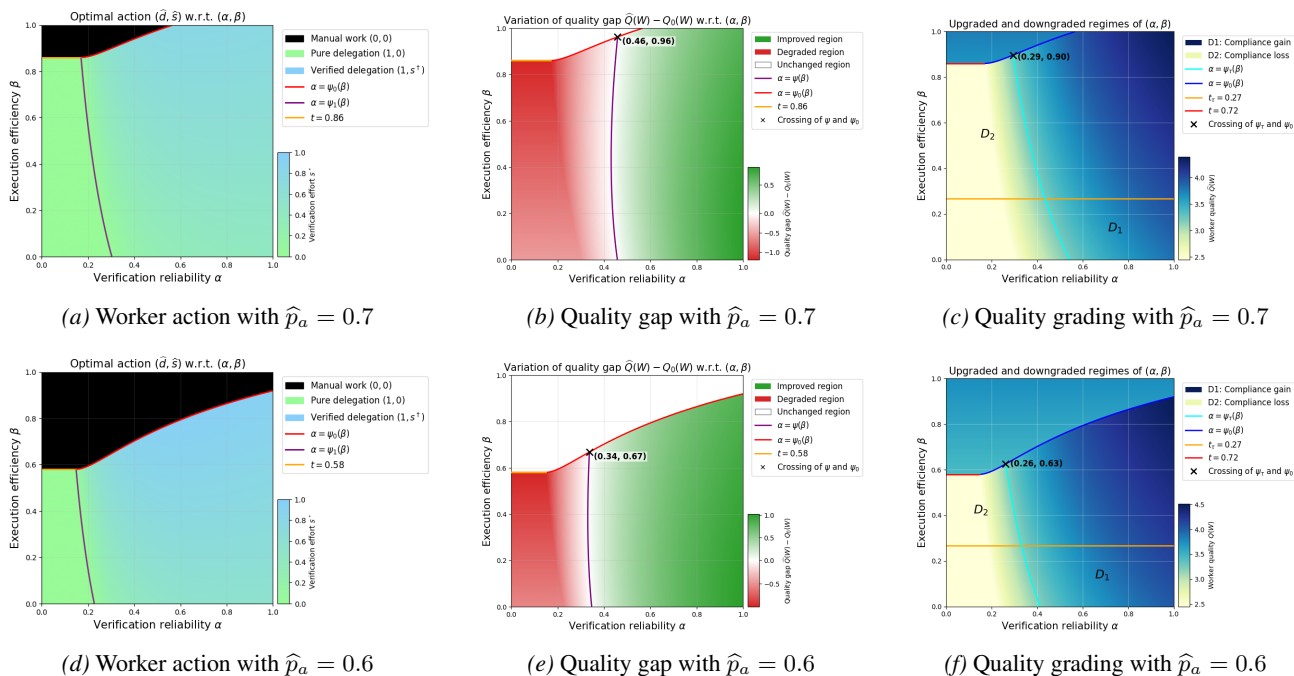

*(a)* Worker action with $\widehat{p}_a = 0.7$      *(b)* Quality gap with $\widehat{p}_a = 0.7$      *(c)* Quality grading with $\widehat{p}_a = 0.7$

*(d)* Worker action with $\widehat{p}_a = 0.6$      *(e)* Quality gap with $\widehat{p}_a = 0.6$      *(f)* Quality grading with $\widehat{p}_a = 0.6$

*Figure 8.* Plots illustrating the worker action and quality regimes under miscalibrated beliefs about AI capability as functions of abilities $(\alpha, \beta)$. The first row reports over-estimation of AI capability, while the second row reports under-estimation of AI capability. The default parameter setting is $(b_W, \ell_W, b_I, \ell_I, \xi, \tau, p_a, C_a, p_w) = (8, 6, 14, 12, 0.3, 6.4, 0.65, 0, 0.75)$, with the functional choices $C_v(s) = s$, $C_w(\beta) = 5(1 - \beta)$, and $\phi(s; \alpha) = 1 - \frac{1}{1 + 2\alpha s}$.

This perturbation preserves the same payoff structure as the baseline model but changes which workflows are attainable. In particular, manual work is feasible only when $C_w(\beta) \leq \tau_c$. At the same time, because verification also consumes resources, the constraint can make high verification effort infeasible. Hence, when the resource budget is tight, workers with low verification reliability may be pushed toward pure delegation rather than verified delegation.

**Analysis.** Figure 7 visualizes the worker action and quality regimes under this perturbation with $\tau_c = 0.6$. Compared with the baseline, a subset of workers originally located in the manual-work regime now switches to pure delegation (Figure 7a). Although these workers would otherwise avoid AI, the resource constraint makes manual execution infeasible while low $\alpha$ makes verification unattractive. Consequently, the quality-degradation region and the compliance-loss region expand in the corresponding area (Figures 7b and 7c). This illustrates a feasibility-driven form of rational over-delegation: workers rely on AI not because delegation is institutionally beneficial, but because their feasible non-AI workflows are constrained.

Overall, the persistence of phase transitions, quality degradation, and compliance loss under this perturbation supports the robustness of our main conclusions.

### C.3. Belief miscalibration and reliance behavior

In the baseline model, the worker is assumed to know the true AI success probability $p_a$. In practice, however, AI performance is often jagged across instances (Dell'Acqua et al., 2023). As a result, the worker may optimize according to a subjective belief $\widehat{p}_a$ that differs from the true AI capability. This belief perturbation changes the worker's chosen action, while the realized institutional quality is still evaluated under the true $p_a$.

We distinguish two forms of belief miscalibration. The first is overestimation, where $\widehat{p}_a > p_a$, which may induce over-reliance on AI. The second is underestimation, where $\widehat{p}_a < p_a$, which may induce under-reliance.

Let $\widehat{p}_a \in [0, 1]$ denote the success probability that the worker believes the AI to have. This miscalibrated belief modifies the worker utility to

$$\widehat{U}_W(d, s) = \big[(1 - \widehat{p}_a)\big((b_W + \ell_W)p_w - C_w\big)\phi(s) - C_v(s) - (b_W + \ell_W)(p_w - \widehat{p}_a) + C_w - C_a\big]d + g_W.$$

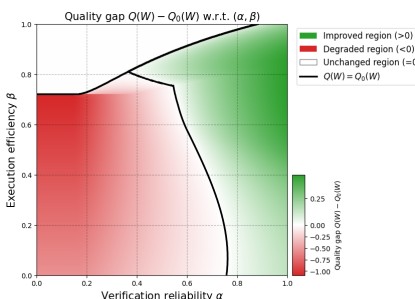
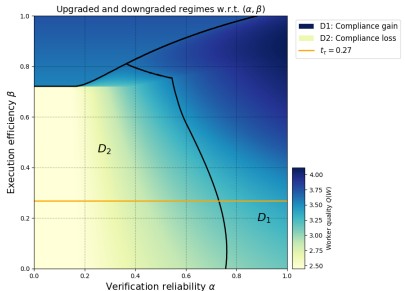

*(a)* Quality gap with heterogeneous task difficulty

*(b)* Quality grading with heterogeneous task difficulty

*Figure 9.* Plots illustrating the worker quality regimes with heterogeneous task difficulty as functions of abilities $(\alpha, \beta)$, under the default parameter setting $(b_W, \ell_W, b_I, \ell_I, \xi, \tau, C_a) = (8, 6, 14, 12, 0.3, 6.4, 0)$, and the functional choices $p_w(h) = 1 - 0.5h$, $p_a(h) = 1 - 0.7h$, $C_v(s; h) = (0.5 + h)s$, $C_w(\beta; h) = 10(1 - \beta)h$, and $\phi(s; \alpha) = 1 - \frac{1}{1 + 2\alpha s}$.

Let

$$(\widehat{d}, \widehat{s}) \in \arg \max_{(d,s) \in [0,1]^2} \widehat{U}_W(d, s)$$

denote the worker's optimal action under the miscalibrated utility. The resulting worker quality is

$$\widehat{Q}(W) := U_I(\widehat{d}, \widehat{s}).$$

In particular, $\widehat{Q}(W) = Q(W)$ when $\widehat{p}_a = p_a$.

**Overestimation and over-reliance: $\widehat{p}_a > p_a$.** Figures 8a–8c visualize how worker action and worker quality vary when the worker overestimates AI capability, with $\widehat{p}_a = 0.7 > p_a = 0.65$. The core qualitative phenomena from the baseline model persist, including phase transitions, compliance gain/loss regimes, and rational over-delegation. Moreover, over-delegation becomes more pronounced, expanding both the quality-degradation region and the compliance-loss region. This highlights the risk of overestimating AI capability and the importance of calibrating workers' beliefs about AI performance.

**Underestimation and under-reliance: $\widehat{p}_a < p_a$.** Conversely, Figures 8d–8f consider the opposite form of belief miscalibration, where the worker underestimates AI capability, with $\widehat{p}_a = 0.6 < p_a = 0.65$. This captures an under-reliance regime: pessimistic beliefs reduce the perceived return from delegation, causing workers to rely less on AI and expanding the manual-work region. Compared with the calibrated benchmark, both the improved and degraded quality regions shrink. Thus, under-reliance dampens the overall effect of AI assistance: it can reduce harmful over-delegation, but it can also prevent workers from realizing potential gains from AI access.

Together, these two belief perturbations show that whether workers over-rely on or under-rely on AI, the core phase-transition structure and heterogeneous quality effects remain intact, demonstrating the robustness of our model and conclusions.

### C.4. Heterogeneous task difficulty

Our model implicitly assumes that tasks have uniform difficulty, leading to constant success probabilities and costs. In practice, workers may face tasks of varying difficulty; for example, clinical cases can differ substantially in diagnostic hardness. To study the effect of heterogeneous task difficulty, we extend the model as follows.

Let $h \in [0, 1]$ denote task difficulty, where $h = 0$ is easiest and $h = 1$ is hardest. We model success probabilities as

$$p_w(h) = 1 - 0.5h, \qquad p_a(h) = 1 - 0.7h$$

for the worker and the AI, respectively. Both $p_w(h)$ and $p_a(h)$ are monotonically decreasing in $h$. This specification keeps the AI-alone success probability below the worker-alone success probability, as in the baseline calibration. Thus, the value of AI in this example does not come from superior standalone accuracy, but from changing the cost and workflow tradeoff faced by the worker.

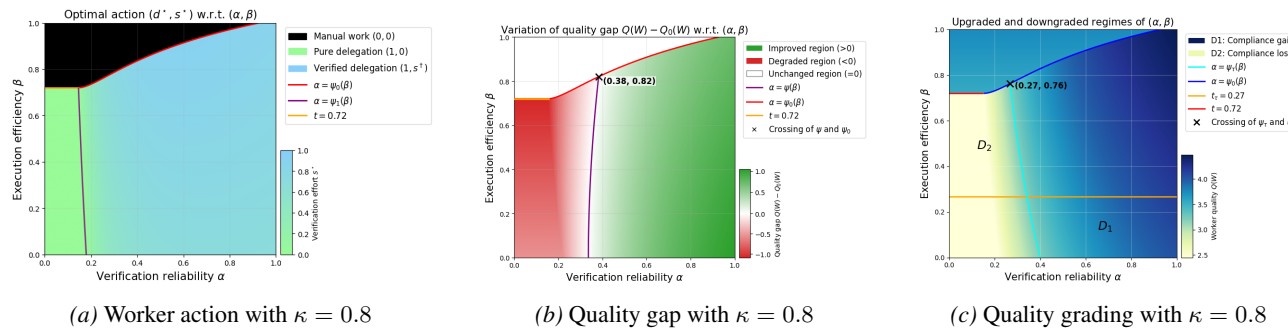

*(a)* Worker action with $\kappa = 0.8$        *(b)* Quality gap with $\kappa = 0.8$        *(c)* Quality grading with $\kappa = 0.8$

*Figure 10.* Plots illustrating the worker action and quality regimes with partial re-execution cost as functions of abilities $(\alpha, \beta)$, under the default parameter setting $(b_W, \ell_W, b_I, \ell_I, \xi, \tau, p_a, C_a, p_w, \kappa) = (8, 6, 14, 12, 0.3, 6.4, 0.65, 0, 0.75, 0.8)$, and the functional choices $C_v(s) = s$, $C_w(\beta) = 5(1 - \beta)$, and $\phi(s; \alpha) = 1 - \frac{1}{1 + 2\alpha s}$.

We model the execution cost and verification cost as

$$C_w(\beta; h) = 10(1 - \beta)h, \qquad C_v(s; h) = (0.5 + h)s,$$

respectively, both of which are monotonically increasing in $h$ for any fixed $s$. All other parameters and functional forms are set as in Figure 1. In particular, when $h \equiv 0.5$, this specification reduces to the setting of Figure 1.

For each difficulty level $h$, we compute the worker's optimal action $(d^\star(h), s^\star(h))$ and the resulting quality $Q(W; h)$. Let the task-difficulty distribution be $\mu = \text{Unif}[0, 1]$. The expected worker quality under heterogeneous task difficulty is then

$$Q(W) = \int_0^1 Q(W; h)\, dh.$$

Analogous to Section 3, we study quality variation and grading outcomes by comparing $Q(W)$ with the no-AI baseline $Q_0(W)$.

**Analysis.** Figure 9 visualizes how worker quality varies under AI assistance when task difficulty is heterogeneous. Averaging over task difficulty smooths the regime boundaries relative to the fixed-difficulty benchmark, because workers may choose different workflows at different difficulty levels. Nevertheless, the aggregate quality gap still contains both improvement and degradation regions, and the grading plot still exhibits compliance gain and compliance loss. Thus, task heterogeneity does not wash out the baseline mechanism.

## C.5. Partial re-execution

Our model assumes that after detecting an AI error, the worker fully re-executes the task and incurs the full cost $C_w$. In practice, re-execution may be only partial, since the worker can often leverage the AI output, for example in essay writing or report generation, to reduce effort.

Let $\kappa \in [0, 1]$ denote a re-execution cost factor, where $\kappa = 1$ corresponds to full re-execution as in the baseline model, and $\kappa < 1$ captures partial re-execution. Under this extension, the worker utility becomes

$$U_W(d, s) = \left[ (1 - p_a)\big((b_W + \ell_W)p_w - \kappa C_w\big)\phi(s) - C_v(s) - (b_W + \ell_W)(p_w - p_a) + C_w - C_a \right] d + g_W.$$

This extension changes the cost of correcting detected AI errors, while leaving the delegation and verification structure otherwise unchanged.

**Analysis.** Figure 10 visualizes how the worker's action and quality vary under partial re-execution cost with $\kappa = 0.8$. The core qualitative phenomena from Figure 1 persist, including phase transitions, compliance gain/loss regimes, and rational over-delegation. Notably, Figure 10 is nearly identical to Figure 1, suggesting that partial re-execution has only a mild effect on worker quality in this regime. The reason is that partial re-execution only changes the marginal cost of correcting detected errors, while the basic tradeoff between delegation surplus and verification reliability remains unchanged.

## C.6. Multi-round interaction

We now relax the assumption that verification occurs in a single step. Under delegation, the worker may interact with the AI over multiple rounds before accepting, revising, or replacing the AI output. Suppose the delegated workflow consists of $T$ verification rounds. In each round $t \in \{1, \ldots, T\}$, the worker exerts verification effort $e_t \geq 0$, which detects an AI error with probability $\delta_t(e_t) \in [0, 1]$. Conditional on the effort sequence $e = (e_1, \ldots, e_T)$, we assume that detection events are independent across rounds. The probability that an incorrect AI output is detected by the end of the interaction is therefore

$$\phi_T(e_1, \ldots, e_T) := 1 - \prod_{t=1}^{T} \big(1 - \delta_t(e_t)\big).$$

If an error is detected, the worker re-executes the task independently with success probability $p_w$. Hence, under delegation, the success probability is

$$p_T^{\text{del}}(e_1, \ldots, e_T) = p_a + (1 - p_a)\phi_T(e_1, \ldots, e_T)p_w,$$

and the overall success probability under delegation rate $d$ is

$$p_T(d, e_1, \ldots, e_T) = (1 - d)p_w + dp_a + d(1 - p_a)\phi_T(e_1, \ldots, e_T)p_w.$$

Let the multi-round verification cost be

$$C_{v,T}(e_1, \ldots, e_T) := \sum_{t=1}^{T} c_t(e_t),$$

where $c_t(\cdot)$ denotes the cost of effort in round $t$. The induced total cost is

$$\text{cost}_T(d, e_1, \ldots, e_T) = (1 - d)C_w + d\Big(C_a + C_{v,T}(e_1, \ldots, e_T) + (1 - p_a)\phi_T(e_1, \ldots, e_T)C_w\Big).$$

This multi-round perturbation can be represented as a reduced-form one-shot verification technology. To see this, define the effective detection frontier under total verification effort $s$ as

$$\bar{\phi}_T(s) := \max_{\substack{e_1, \ldots, e_T \geq 0 \\ \sum_{t=1}^{T} e_t \leq s}} \left\{ 1 - \prod_{t=1}^{T} \big(1 - \delta_t(e_t)\big) \right\}.$$

Let $e^\star(s)$ denote an effort allocation attaining this maximum, and define the induced effective verification cost by

$$\bar{C}_{v,T}(s) := \sum_{t=1}^{T} c_t(e_t^\star(s)).$$

Substituting these effective functions into the expressions above yields

$$p_T(d, s) = (1 - d)p_w + dp_a + d(1 - p_a)\bar{\phi}_T(s)p_w,$$

and

$$\text{cost}_T(d, s) = (1 - d)C_w + d\Big(C_a + \bar{C}_{v,T}(s) + (1 - p_a)\bar{\phi}_T(s)C_w\Big).$$

Thus, the multi-round interaction has the same structural form as the baseline model, with the original detection function $\phi$ and verification cost $C_v$ replaced by the effective functions $\bar{\phi}_T$ and $\bar{C}_{v,T}$.

For example, in the homogeneous case with equal effort allocation $e_t = s/T$ and exponential per-round detection $\delta_t(e) = 1 - e^{-\alpha e}$, we obtain

$$\bar{\phi}_T(s) = 1 - \left(e^{-\alpha s/T}\right)^T = 1 - e^{-\alpha s},$$

which coincides with the detection function used in the baseline specification.

This reduction shows that the qualitative conclusions of the baseline analysis are preserved under multi-round interaction. Multi-round verification may change the effective detection frontier and the cost of scrutiny, but worker quality continues to be governed by the interaction between delegation incentives and verification reliability.

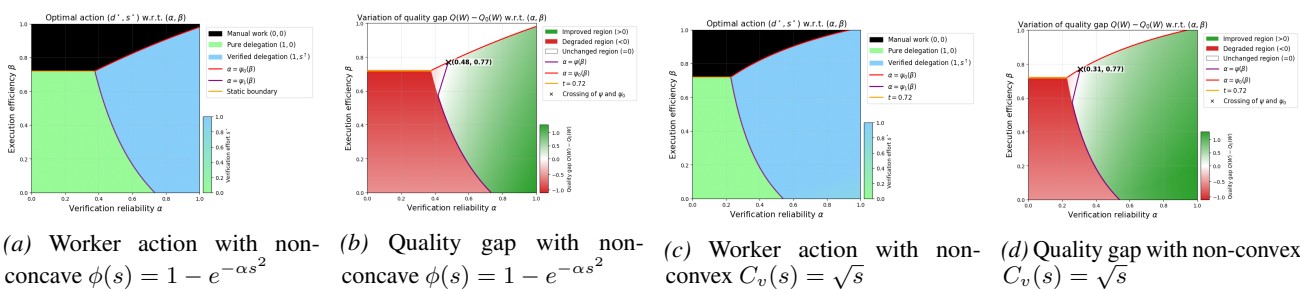

*(a)* Worker action with non-concave $\phi(s) = 1 - e^{-\alpha s^2}$

*(b)* Quality gap with non-concave $\phi(s) = 1 - e^{-\alpha s^2}$

*(c)* Worker action with non-convex $C_v(s) = \sqrt{s}$

*(d)* Quality gap with non-convex $C_v(s) = \sqrt{s}$

*Figure 11.* Plots illustrating the worker action and quality regimes under relaxed regularity assumptions for the detection function $\phi(\cdot)$ and verification cost $C_v(\cdot)$, as functions of abilities $(\alpha, \beta)$. The left two panels use the non-concave detection function $\phi(s; \alpha) = 1 - e^{-\alpha s^2}$, while the right two panels use the non-convex verification cost $C_v(s) = \sqrt{s}$. Other parameters are as in Figure 1.

## C.7. Continuous output quality

The baseline model evaluates task outcomes through a binary success indicator. This abstraction is natural for tasks with objectively correct labels, but many AI-assisted outputs are better described by a continuous quality score: a diagnosis, legal memo, or generated report may be partially correct rather than simply correct or incorrect. To capture this setting, let $Y(d, s) \in [0, 1]$ denote the final output quality induced by delegation level $d$ and verification effort $s$, and let $Y_w$ and $Y_a$ denote the worker-alone and AI-alone output qualities. Write

$$q_w := \mathbb{E}[Y_w], \qquad q_a := \mathbb{E}[Y_a].$$

We incorporate continuous quality by applying the same detection–correction logic as in the baseline model. Verification detects the residual quality deficit of the AI output with probability $\phi(s)$. A detected deficit is then partially repaired by the worker. In this formulation, $q_w$ is interpreted as the worker's ability to repair the residual quality deficit of the AI output, rather than as simple replacement of the AI output by an independently produced worker output. The expected output quality is therefore

$$p_Y(d, s) := \mathbb{E}[Y(d, s)] = (1 - d)q_w + d\big(q_a + (1 - q_a)\phi(s)q_w\big).$$

Thus, the only substantive change is to replace the binary success probabilities $p_w$ and $p_a$ with the expected quality levels $q_w$ and $q_a$, and to replace the AI error probability $1 - p_a$ with the expected residual quality gap $1 - q_a$. When $Y_w$ and $Y_a$ are Bernoulli random variables, this expression reduces exactly to the baseline success probability.

The correction-cost term is modified analogously by replacing $1 - p_a$ with $1 - q_a$, while the definitions of worker utility, institutional utility, optimal action, and worker quality remain unchanged. Because these substitutions preserve the same reduced-form utility structure as the baseline model, all subsequent results apply directly under the continuous-quality interpretation, including the characterization of worker actions, quality improvement and degradation, compliance gain/loss regimes, and rational over-delegation.

Thus, the continuous-quality extension shows that our framework is not tied to binary correctness and further supports the robustness of the model and conclusions.

## C.8. Relaxing concavity and convexity assumptions

The baseline model imposes two regularity assumptions on the verification technology: the detection function $\phi(s; \alpha)$ is concave in verification effort, and the verification cost $C_v(s)$ is convex. These assumptions make the worker's verification problem well behaved, but they are not essential for the model's main mechanism. In this subsection, we relax them separately.

**Non-concave detection function.** We first relax the concavity of $\phi$ while keeping the baseline verification cost. This relaxation may destroy the concavity of the worker's verification objective, so the optimal verification effort $s^\dagger \in \arg\max_{s \in [0,1]} f_W(s)$ need not be unique or continuous. However, the main structural property is preserved: since $U_W(d, s) = f_W(s)d + g_W$ remains affine in $d$, the delegation decision is still governed by whether the maximized delegation surplus $\max_s f_W(s)$ is nonnegative. Provided the pointwise monotonicity of the delegation surplus in $\alpha$ and $\beta$ is preserved,

the maximized surplus $\max_s f_W(s)$ inherits the same monotone comparative statics. Hence the threshold structure behind the phase transition persists even without concavity.

Figures 11a and 11b illustrate this case with the non-concave detection function $\phi(s;\alpha) = 1 - e^{-\alpha s^2}$. The same manual-work, pure-delegation, and verified-delegation regimes remain, and the quality-gap plot still contains a low-$\alpha$ degradation region. The main new feature is that verification has weak marginal value near $s = 0$, so as $\alpha$ increases the global maximizer can jump discretely from $s^\star = 0$ to a positive verification level. This creates turning points in the regime boundary, but it does not eliminate rational over-delegation or the associated quality-degradation region.

**Non-convex verification cost.** We next relax the convexity of $C_v$ while keeping the baseline concave detection function. The same qualitative logic applies: the verification objective may lose smoothness and concavity, but the worker's delegation choice is still determined by the threshold in $\max_s f_W(s)$.

Figures 11c and 11d consider the non-convex cost $C_v(s) = \sqrt{s}$. Because small positive verification effort is relatively costly near the origin, the worker again avoids infinitesimal verification and switches to positive verification only when the global verification surplus becomes sufficiently large. The resulting action and quality-gap plots mirror the non-concave-$\phi$ case: regime boundaries become less smooth and verification effort may jump, but the core regions—manual work, pure delegation, verified delegation, and low-$\alpha$ quality degradation—remain present.

Overall, relaxing the concavity of $\phi$ or the convexity of $C_v$ changes the smoothness of verification choices but not the core mechanism, showing that our main conclusions are robust to these regularity perturbations.

Taken together, these extensions show that the baseline conclusions are robust across changes in the action space, information structure, task environment, correction workflow, outcome representation, and regularity assumptions. The extensions modify the geometry of regime boundaries and the size of gain/loss regions, but they do not remove the core mechanism: delegation is beneficial only when paired with sufficiently reliable verification, while rational private delegation can generate institutional quality loss when verification is weak.

## D. Omitted Details in Section 5: From Data to Model Parameters

We provide additional details for the example stated in Section 5.

**Collab-CXR dataset.** The Collab-CXR dataset is organized into multiple experimental designs. We focus on diagnoses collected under **Design 2**, in which each clinician is randomly assigned 60 cases out of the 324 total cases. For each case, the clinician makes predictions under one of four information environments: (1) X-ray only; (2) X-ray + AI-predicted probabilities; (3) X-ray + medical history; and (4) X-ray + medical history + AI-predicted probabilities. The study is conducted over four sessions separated by at least two weeks. In each session, the clinician diagnoses cases under exactly one of the four information environments, and across the four sessions each of the 60 cases is eventually observed under all four environments. This design enables direct comparisons of clinician performance with and without AI assistance.

Since the AI diagnoses cases using only the X-ray, we focus on information environments (1) and (2) to enable a fair comparison between the AI and clinicians. Accordingly, we treat environment (1) as the clinician-alone condition and environment (2) as the clinician-with-AI condition.

**Details of case success.** Given a case $j$, let $X_a^{(j)}, X_w^{(j)} \in \{0,1\}^{14}$ denote the AI's and the clinician's diagnosis-indicator vectors, respectively. The $i$-th coordinate equals 1 if the prediction for the $i$-th pathology matches the expert label. Let $I_a^{(j)}, I_w^{(j)} \in \{0,1\}$ denote the corresponding *case-success* indicators for the AI and the clinician. We define

$$I_a^{(j)} = \prod_{i \in [14]} (X_a^{(j)})_i, \qquad I_w^{(j)} = \prod_{i \in [14]} (X_w^{(j)})_i.$$

**Case clean.** Case difficulty may vary, which can induce substantial variation in the clinician-alone time cost $C_w$. To restrict attention to cases of comparable hardness, we retain only those cases whose clinician-alone completion time lies within $[0.5\times, 2\times]$ the clinician's average completion time.

In addition, there are a small number of cases $j$ with $I_a^{(j)} = 1$ but $I_w^{(j)} = 0$, i.e., cases where the clinician appears to override an AI output that matches the expert labels. This behavior is not captured by our baseline model, which assumes clinicians

only detect AI errors. Since such cases account for only $4.9\%$ of all cases, we treat them as erroneous-detection cases and exclude them from the analysis.

After this data cleaning, we retain 38 cases for clinician `vn_exp_65341958`.

**Details of deriving** $\mathrm{cost}(1, s^\dagger)$. We observe that the average time spent in the clinician-with-AI condition, denoted $C_{wa}$, is larger than the clinician-alone average time $C_w$. This is because recorded time includes not only diagnostic reasoning but also time spent interacting with the interface (e.g., reviewing the AI output and entering information). In particular, even when the clinician does not modify the AI output (captured by $X_w = X_a$), there is still nontrivial interface/entry time.

We approximate this fixed entry-time component by $C_w$ and use the decomposition

$$C_{wa} = \mathrm{cost}(1, s^\dagger) + \Pr[X_w = X_a] \cdot C_w.$$

Consequently, we estimate

$$\mathrm{cost}(1, s^\dagger) = C_{wa} - \Pr[X_w = X_a] \cdot C_w.$$

**Parameter categorization.** The parameters used in the empirical illustration fall into three categories. First, some quantities are directly observed or estimated from outcome data, including the worker's baseline success probability $p_w$, the AI's standalone success probability $p_a$, and the observed success rate under delegation $p(1, s^\dagger)$. Second, worker-side quantities are inferred through the model by matching observed behavior and outcomes. These include the verification choice $s^\dagger$, the effective detection probability $\phi(s^\dagger)$, the verification cost $C_v(s^\dagger)$, composite cost terms such as $\mathrm{cost}(1, s^\dagger)$, the ability parameters $\alpha$ and $\beta$, and the net private payoff scale $b_W + \ell_W$. Third, structural parameters that are not identified from worker-level observational data are fixed by assumption for counterfactual evaluation. These include the AI usage cost $C_a$, the institutional benefit and loss parameters $b_I$ and $\ell_I$, the cost-discount factor $\xi$, and the institutional quality threshold $\tau$. This categorization clarifies the role of the empirical exercise: it maps available data into the model's primitives and illustrates operational implications, rather than providing a causal identification of all structural parameters.

