# OpenReview forum: "Delegation and Verification under AI"
_ICML.cc/2026/Conference — ICML 2026 regular_

### Official Review · Reviewer_qSKD · 2026-03-10

**Soundness:** 3
**Presentation:** 4
**Significance:** 3
**Originality:** 4
**Overall Recommendation:** 5
**Confidence:** 2

**Summary:**

The paper at hand deals with an interesting and timely topic: It simulates institutional workflows as a utility-driven OR problem with a specific focus on worker's ability to either execute a task manually, delegate it to AI or delegate it to AI and verify the output. The results show that when playing through different scenarios, so-called "phase transitions" can be observed, where different phases bring different pros and cons with them. The paper shows these transitions both theoretically but also with a real-world data set. In conclusion, the authors show that the introduction of AI assistance in organizational workflows reshapes the quality of the workers in different ways, depending on their ability to appropriately rely & verify AI output.

**Compliance With Llm Reviewing Policy:**

Affirmed.

**Final Justification:**

After the reviewing and rebuttal period, I still believe this is a strong paper and I stick to my initial judgement of an Accept. All concerns have been addressed adequately.

**Key Questions For Authors:**

- How would you model more nuanced reliance behaviours on the worker's side, focussing more on aspects of time/cognitive effort/expertise?
- How would this work look like if we have no binary choice of "correctness", but more nuanced aspects (like it is common in the GenAI world)?
- What can organizations learn from the work?
- Can you justify each choice of the parameters more soundly in the appendix?

**Limitations:**

yes

**Strengths And Weaknesses:**

The work at hand shows many strengths, which I will list below.

A) The topic is very timely, and the authors do a great job in translating real-world organizational workflows into a theoretical model with all necessary variables and circumstances in place. This abstraction is far from trivial.

B) The simulation is extensive and performed both theoretically as well as with a real-world data set.

C) The results are meaningful and inform the dynamics of workforces in an organization. It captures many discussions on reliance behavior and organizational transformation that are currently discussed in both scientific and journalistic work.

However, there are also multiple shortcomings that need to be addressed:

D) Parameterization is everything in these experiments. The authors should add additional robustness checks in their appendix, i.e., model different worker utilities apart from the ones mentioned.

E) The body of literature on verification and reliance is more complex than depicted in the manuscript. The underlying assumption is over-reliance and oversight decay, but the empirical results in the field are more nuanced. This is also something that should be considered in the model: What if a fundamental share of the workers rather under-relies than over-relies?

F) The authors should theorize more on the relationship between verification (ability) and accountability; this aspect is only touched upon briefly.

G) I would urge the authors to tone down some aspects, e.g. "Consequently, AI induces a nonlinear and discontinuous mappig from ability to action." --> Yes, bc it was modeled this way!

Overall, I feel the general work adds a meaningful delta to the state-of-the-art. Whether it would be better suited at an ECON outlet or the ICML is hard for me to judge.

---

> ### Author Rebuttal · Authors · 2026-03-31
>
> We thank you for your encouraging review. We are especially grateful for your recognition of the novelty of the topic and for your thoughtful suggestions regarding more nuanced reliance behaviors, which we have now incorporated. Please see this PDF for new figures (link: https://acrobat.adobe.com/id/urn:aaid:sc:eu:2f91c039-23cc-4998-8ab4-9e03569a3508).
>
> We will add additional robustness checks (including alternative utility specifications) and revise statements to clarify their dependence on model assumptions in the final version.
>
> >*“How would you model more nuanced reliance behaviours on the worker's side, focussing more on aspects of time/cognitive effort/expertise?”*
>
> To capture more nuanced reliance behaviours, we introduce a budget $\tau_c$ on the worker's total execution and verification effort:
> $$(1-d)C_w+dC_v(s)\leq\tau_c.$$
> This capacity constraint restricts feasible verification, lowering the optimal effort $s^\dagger$ and forcing heavier reliance on AI. As $\tau_c$ decreases, workers originally in the manual-work regime are forced into pure delegation (Fig. 2(a) in the PDF). Consequently, the institutional utility $U_I$ derived from these workers drops, further exacerbating rational over-delegation (Figs. 2(b) and 2\(c) in the PDF).
>
> We will add this extension to the final version.
>
> >*“How would this work look like if we have no binary choice of "correctness", but more nuanced aspects”*
>
> We agree that task outcomes can fall on a continuous quality scale rather than being strictly binary. Below we show that our framework captures this.
>
> Let continuous output quality be $Y(d,s)\in [0,1]$, with worker and AI baselines $Y_w$ and $Y_a$. Assuming verification improves quality analogously to the binary case, taking the expectation $p(d,s):=\mathbb{E}[Y(d,s)]$ yields:
> $$p(d,s)=(1-d)\mathbb{E}[Y_w]+d(\mathbb{E}[Y_a]+(1-\mathbb{E}[Y_a])\phi(s)\mathbb{E}[Y_w]).$$
>
> Thus the reduced-form structure is preserved. We will include this continuous-outcome extension in the appendix of the final version.
>
> >*“Can you justify each choice of the parameters more soundly in the appendix?”*
>
> Thank you. We will add an appendix discussion explaining our parameter choices.
>
> We choose parameters satisfying the natural ordering $b_I>b_W$, $\ell_I>\ell_W$, and $\xi<1$ to reflect that institutions face higher stakes while bearing only fractional private costs ($b_W=8$, $\ell_W=6$, $b_I=14$, $\ell_I=12$, $\xi=0.3$). We set $p_w=0.75>p_a=0.65$ to model high-risk settings in which humans remain more accurate than moderately reliable but cheaper AI. We use simple linear functional forms for interpretability, since our results depend only on structural properties rather than specific functional choices (Remark 2.1). Finally, $\tau=6.4$ makes the pre-AI baseline $Q_0(W;\beta)=6+1.5\beta$ screen out only a small fraction of unqualified workers.
>
> >*“What can organizations learn from the work?”*
>
> Our results offer concrete lessons for organizations deploying AI:
>
> **Better AI can worsen outcomes:** More capable AI expands the quality-degradation regime for workers with low verification reliability (Thm. 3.5).
>
> **Monitor workflows, not just results:** Outcome-based evaluation alone misses individually rational but institutionally harmful workers in the compliance-loss regime (Thm. 3.6).
>
> **Interventions are non-monotonic:** Increasing AI capability or reshaping incentives improves quality for some but degrades it for others, demanding heterogeneity-aware policies (Sec. B.2).
>
> We will add an implications section in the final version.
>
> >*“...What if a fundamental share of the workers rather under-relies than over-relies?”*
>
> We agree that under-reliance is an important real-world phenomenon, and our framework can capture it.
>
> In our model, under-reliance arises when workers underestimate AI capability, i.e., $\widehat{p}_a<p_a$. This naturally allows for heterogeneous populations in which some workers persistently under-rely on AI.
>
> Simulations of worker actions and institutional quality under $\widehat{p}_a<p_a$ show that such pessimistic beliefs expand the manual-work regime (Fig. 5(a) in the PDF), as workers rationally reduce AI reliance.
>
> We will include this analysis in the final version.
>
> >*“...theorize more on the relationship between verification (ability) and accountability...”*
>
> In our framework, accountability is not an additional primitive, but an equilibrium property of the delegation--verification pipeline. Once execution is delegated, a worker can only influence the final outcome through the correction channel $d(1-p_a)\phi(s)p_w$, meaning substantive accountability requires both detecting an AI error and successfully re-executing the task. Therefore, while formal responsibility may be assigned to the human, true accountability is endogenous, existing only when the worker's verification ability makes oversight theoretically feasible and incentive-compatible.
>
> We will revise the paper to make this interpretation explicit.

---

> > ### Author Rebuttal · Reviewer_qSKD · 2026-04-01
> >
> > My concerns have been fully addressed — as far as this is possible w/o consulting the actual revision.

---

### Official Review · Reviewer_BUor · 2026-03-12

**Soundness:** 4
**Presentation:** 3
**Significance:** 4
**Originality:** 3
**Overall Recommendation:** 4
**Confidence:** 3

**Summary:**

This paper presents a utility-driven framework to model how AI assistance impacts human workflows and reshapes institutional worker quality. It models human-AI interaction as a delegation and verification pipeline, where workers optimize delegation probability and verification effort to maximize personal utility. The paper demonstrates that individually rational delegation strategies can fail to maximize institutional utility due to misaligned evaluation standards. The primary finding is that AI introduces non-linear phase transitions (or sometimes even "bipolar") in behavior, amplifying the value of workers with strong verification skills while degrading institutional quality for those with weaker skills who over-delegate.

**Compliance With Llm Reviewing Policy:**

Affirmed.

**Key Questions For Authors:**

* In practice, a worker's interaction with AI is more than two rounds. Usually, it involves a dynamic process where the worker adjusts the prompt or material he/she supplements the AI tool based on the AI's response or request. Does your model capture this situation, or if not, is there a simple way to extend the current result to this dynamic situation?
* As stated before, the work lacks sufficient study of the equilibrium outcome between the workers and the institution. How would you expect the equilibrium outcome if both workers and institutions are strategic?
* Can you give an example of why the cost of verification effort has an increasing margin in the AI verification context?

**Limitations:**

yes

**Strengths And Weaknesses:**

The submission is technically sound. The modeling is rigorous and consistent with microeconomics principles. The presentation flow is clear and well-structured. The significance of the work lies in the fact that the authors theoretically proved humans' over-reliance on AI is a rational behavior rather than a cognitive load or psychological burden. Also, this work innovatively, up to the best of my knowledge, models the workers' verification effort as an endogenous choice.

However, the authors only consider the strategic action of one stakeholder---the worker in the workspace. It would be a plus if the authors studied the equilibrium behavior between strategic interactions between the worker and the institution on the AI adoption.

---

> ### Author Rebuttal · Authors · 2026-03-31
>
> Thank you for your valuable and constructive feedback. We appreciate your recognition of model novelty and the suggestion regarding multi-round interaction and stategic institution, which we have now incorporated. Please see this PDF for new figures (link: https://acrobat.adobe.com/id/urn:aaid:sc:eu:2f91c039-23cc-4998-8ab4-9e03569a3508).
>
> >*“In practice, a worker's interaction with AI is more than two rounds...Does your model capture this situation, or if not, is there a simple way to extend the current result to this dynamic situation?”*
>
> Thank you for the insightful question. We agree that capturing the dynamic, multi-round nature of prompting and refinement is a valuable extension. While our baseline model abstracts worker-AI interaction into a single step, it can be extended to capture multi-round dynamic interactions by aggregating effort.
>
> Consider a worker interacting with the AI over $T$ rounds. In each round $t$, the worker exerts effort $s_t$, detecting errors with probability $\delta_t(s_t)$. The cumulative probability of detecting an error across $T$ conditionally independent rounds is:
> $$
> \phi_T(s_1,\dots,s_T)=1-\prod_{t=1}^T(1 - \delta_t(s_t)).
> $$
> The overall task success probability under action $d$ is therefore:
> $$
> p_T(d,s_1,\dots,s_T)=(1-d)p_w+dp_a+d(1-p_a)\phi_T(s_1,\dots,s_T)p_w.
> $$
>
> Optimizing over a fully independent, multi-dimensional effort vector $(s_1, \dots, s_T)$ leads to a dynamic program. However, if we assume a symmetric allocation of a total effort budget $s$ (where $s_t = s/T$ for all $t$), the cumulative detection function simplifies to a single-variable function:
> $$
> \phi_T(s)=1-(1 - \delta(s/T))^T.
> $$
> Substituting this back yields the same reduced-form structure as in the baseline model:
> $$
> p_T(d,s)=(1-d)p_w+dp_a+d(1-p_a)\phi_T(s)p_w.
> $$
> A similar reduction holds for $C_v(s)$ and $\mathrm{cost}(d,s)$.
>
> By replacing the original $\phi(s)$ and $C_v(s)$ with their $T$-round effective counterparts, the structural misalignment and phase-transition analysis extend within the same framework.
>
> We will add this extension in the final version.
>
> >*“...How would you expect the equilibrium outcome if both workers and institutions are strategic?”*
>
>
> Thank you for this insightful question. We agree that modeling a strategic institution is a crucial extension.
>
> To explore this, we formulate a Stackelberg game between the institution and the worker. Building on the model in Section B.2, suppose the institution can strategically transfer a portion of its utility, $D_b\ge 0$, to the worker. The task success benefit thus becomes $b_W+D_b$ for the worker and $b_I-D_b$ for the institution. Let $Q(D_b)$ denote the resulting institutional worker quality under this reshaped incentive. Furthermore, we include a reduced-form term $C(D_b) = \kappa D_b^2$ capturing additional institutional value arising from improved incentive alignment, yielding a total institutional objective of $Q(D_b)+C(D_b)$.
>
> The game unfolds as follows: the institution first selects the optimal transfer $D_b^\star=\arg\max_{D_b \ge 0} Q(D_b)+C(D_b)$, after which the worker takes the utility-maximizing action given the new incentive $b_W + D_b^\star$.
>
> Simulations illustrate that the equilibrium transfer $D_b^\star$ is often positive (Fig. 4(a) in the PDF). This strategic incentive motivates workers to increase their verification effort, leading to improved institutional quality relative to the baseline $Q(W)$, particularly for workers originally in the manual-work or pure-delegation regimes (Fig. 4(b) in the PDF). In many cases, this intervention upgrades workers relative to the Non-AI baseline $Q_0(W)$ (Fig. 4\(c) in the PDF).
>
> We will include this strategic extension in the final version.
>
> >*“Can you give an example of why the cost of verification effort has an increasing margin in the AI verification context?”*
>
>
> Thank you for the question. A natural example is the verification of AI-generated software code. Early verification, such as unit tests and edge-case checks, is relatively cheap and catches many obvious errors. Once these are removed, the remaining errors are typically deeper logical flaws, requiring slower line-by-line review, control/data-flow tracing, and analysis of module interactions. Eliminating the most subtle vulnerabilities may require formal invariants or full correctness proofs.
>
> This creates diminishing returns in detection: the marginal gain in effective detection $\phi(s)$ falls with effort, so each additional unit of verification requires disproportionately more cognitive and computational effort. Hence the verification cost is naturally convex, with $C_v''(s)>0$.
>
> Thus, convex verification costs arise naturally from the structure of error detection in AI-assisted workflows.

---

> > ### Author Rebuttal · Reviewer_BUor · 2026-04-04
> >
> > Thank you for the detailed response; my question has been fully resolved.

---

### Official Review · Reviewer_jRLG · 2026-03-15

**Soundness:** 3
**Presentation:** 2
**Significance:** 3
**Originality:** 3
**Overall Recommendation:** 4
**Confidence:** 3

**Summary:**

The paper studies an interesting problem of delegation and verification with AI assistants. As AI systems become more capable, workers must decide whether to delegate tasks to AI and how much effort to invest in verifying its outputs, while institutions evaluate them based on outcomes that may not reflect their private costs. The paper formalizes this as a worker's optimization problem and shows that AI can induce phase transitions with several interesting and counterintuitive implications. More broadly, the results suggest that AI may systematically reshape worker performance and may amplify differences between workers with different verification abilities.

**Compliance With Llm Reviewing Policy:**

Affirmed.

**Final Justification:**

My question has been resolved, and I view the paper positively overall.

**Key Questions For Authors:**

1. Which parameters are actually identifiable from observational data, and which are only being calibrated? In Section 4, the paper maps observed clinician performance/time data into quantities like $p_w$, costs, etc. That opens the door to asking how likely one can recover latent parameters like verification reliability $\alpha$ rather than just fit them?

2. How robust are the phase-transition and compliance-loss results to relaxing the concavity/convexity assumptions on verification and cost?

**Limitations:**

The authors have adequately discussed the limitations of their work in the last section.

**Strengths And Weaknesses:**

For soundness, the proposed model and parameters capture the core features of the delegation and verification problem, and the proofs appear correct. One potential concern is that many of the theoretical results rely on structural assumptions about the underlying functions, such as convexity of the cost function or concavity of $\phi$, which may not always hold in practice.

Overall, however, the paper studies an important and timely problem: as AI capabilities improve, rational agents may reduce verification effort, which can in turn worsen institutional outcomes even when average task performance increases. The model is simple yet expressive, and it captures several interesting insights that could plausibly arise in practice.

The presentation is somewhat dense in places, but overall it is fairly clear. I found the example in Section 4 particularly helpful for understanding how the model parameters could be estimated from real data.

In terms of originality, I am not fully familiar with all of the related literature, but the paper seems to offer novel insights and helps deepen our understanding of delegation and verification in settings where workers interact with AI-assisted systems.

---

> ### Author Rebuttal · Authors · 2026-03-31
>
> Thank you for your thoughtful and constructive feedback. We appreciate your recognition of our model and technical results and the suggestion regarding relaxing the concavity assumption, which we have now incorporated. Please see this PDF for new figures (link: https://acrobat.adobe.com/id/urn:aaid:sc:eu:2f91c039-23cc-4998-8ab4-9e03569a3508).
>
> >*“How robust are the phase-transition and compliance-loss results to relaxing the concavity/convexity assumptions on verification and cost?”*
>
> Thank you for the insightful question. We address robustness by relaxing the concavity of the detection function $\phi$ and the convexity of the verification cost $C_v$.
>
> Mathematically, the key object is $f_W(s^\dagger)$. We can show that, under mild regularity conditions, the maximum worker utility under delegation $f_W(s^\dagger)$ remains monotonically increasing in $\alpha$ and decreasing in $\beta$, even when $\phi$ is non-concave and $C_v$ is non-convex.
>
> Since the optimal delegation action follows the threshold rule $d^\star=\mathbb{I}[f_W(s^\dagger)\ge 0]$ (Line 346), this implies that $d^\star$ remains monotonically increasing in $\alpha$ and decreasing in $\beta$. Consequently, the phase transition in $d^\star$ persists. Since compliance loss is driven by rational over-delegation in the pure-delegation regime, and this regime persists under the relaxed assumptions, the compliance-loss results are similarly robust at the level of regime structure.
>
> We also simulate the model using a non-concave detection function $\phi(s)=1-e^{-\alpha s^2}$ and a non-convex verification cost $C_v(s)=\sqrt{s}$ (Fig. 6 in the PDF). The simulations illustrate that the phase-transition structure persists.
>
> Interestingly, a novel phase transition emerges: the worker’s action now jumps discretely from pure delegation ($s^\star=0$) to verified delegation ($s^\star\gg 0$) (Fig. 6(a) in the PDF). This arises because relaxing concavity makes $f_W(s)$ bimodal, so that as $\alpha$ increases, the global maximizer $s^\dagger=\arg\max_{s\in[0,1]} f_W(s)$ shifts discontinuously from the boundary at $0$ to the interior peak.
>
> We will add this extension and the corresponding analysis to the final version.
>
>
> >*“Which parameters are actually identifiable from observational data, and which are only being calibrated?”*
>
>
> Thank you for the question. To clarify the empirical strategy in Section 4, we distinguish between observable quantities, inferred parameters, and fixed structural parameters.
>
> **Directly observable quantities:** The baseline task success probabilities and delegated success rates are observed from data, including $p_w$, $p_a$, and $p(1,s^\dagger)$.
>
> **Inferred from calibration:** Worker-side parameters are recovered by matching the model to observed behavior, including the verification choice $s^\dagger$, the effective detection $\phi(s^\dagger)$, the verification cost $C_v(s^\dagger)$, and composite quantities such as $\mathrm{cost}(1,s^\dagger)$, abilities $\alpha$ and $\beta$, and the net private benefit $b_W+\ell_W$.
>
> **Fixed structural parameters:** Institutional parameters and AI costs are not identified from worker-level data and are set exogenously to evaluate institutional outcomes, including $C_a$, $b_I$, $\ell_I$, $\xi$, and the institutional threshold $\tau$.
>
> We will add a brief summary of this categorization in Section 4 of the final version.

---

> > ### Author Rebuttal · Reviewer_jRLG · 2026-04-02
> >
> > Thank you for the detailed response; my question has been fully resolved.

---

### Official Review · Reviewer_AHT4 · 2026-03-16

**Soundness:** 3
**Presentation:** 3
**Significance:** 4
**Originality:** 4
**Overall Recommendation:** 4
**Confidence:** 3

**Summary:**

The paper models AI-assisted work as a worker choosing delegation probability and verification effort under misaligned worker and institutional utilities in a principal agent framework. Worker quality is defined institutionally, not by the worker’s own objective. The main results characterize three regimes: (i) manual work, (ii) pure delegation, and (iii) verified delegation, and derive phase transitions in behavior, along with compliance-gain and compliance-loss regions as a function of verification reliability and execution efficiency. The paper also includes a calibration-style mapping to radiology data to explain how model parameters could be instantiated.

**Compliance With Llm Reviewing Policy:**

Affirmed.

**Key Questions For Authors:**

* To what extent do the phase transitions survive if delegation enters utility nonlinearly, or if partial delegation is intrinsically valuable rather than just a binary corner solution?

* How would the analysis change if institutions could observe workflow choices and directly reward verification rather than only final outcomes?

**Limitations:**

The model is insightful but highly stylized. The conclusions are interesting hypotheses about AI-assisted work, but not yet compelling evidence.

**Strengths And Weaknesses:**

## Strength

The paper's topic is very timely and proposed problem setup sounds novel and interesting. The paper correctly emphasizes verification ability rather than execution speed, which is interesting than standard “AI boosts productivity” narratives in the literature. The institutional and worker utility distinction is also well-motivated.

## Weakness

The main weakness may be that the proposed model is highly stylized, but I think the results and implications therein overcome such the weakness. Technically, the main discontinuity result is less profound than stated as is. Since utility is affine in delegation, binary delegation regimes are almost immediate in a sense that the phase transitions are partly a consequence of this choice rather than an emergent phenomenon. The empirical section is not really validation but it is closer to a toy calibration on one dataset and one clinician. The jump from stylized theory to claims about workplace redesign feels a bit overlaim.too aggressive.

---

> ### Author Rebuttal · Authors · 2026-03-31
>
> Thank you for your thoughtful and valuable feedback. We appreciate your recognition of our novelties and the suggestion about non-linear utility, which we have incorporated. Please see this new PDF for new figures (link: https://acrobat.adobe.com/id/urn:aaid:sc:eu:2f91c039-23cc-4998-8ab4-9e03569a3508).
>
> >*“To what extent do the phase transitions survive if delegation enters utility nonlinearly, or if partial delegation is intrinsically valuable rather than just a binary corner solution?”*
>
> We thank the reviewer for this insightful question. We agree that allowing for partial delegation is valuable. To explore this, we introduce a convex cost of delegation $\delta d^2$, into the worker's utility function to capture the non-linear cognitive or behavioral friction of adjusting workflows. The utility becomes $U_W(d,s)=f_W(s)d+g_W-\delta d^2$. Optimizing this objective yields $$d^\star=\Pi_{[0,1]} \left(\frac{f_W(s^\dagger)}{2\delta}\right),$$ where $\Pi_{[0,1]}$ is the projection onto $[0,1]$.
>
> Hence the binary transition is replaced by three regimes:
> $d^\star=0$ if $f_W(s^\dagger)\le 0$, $0<d^\star<1$ if $0<f_W(s^\dagger)<2\delta$, and $d^\star=1$ if  $f_W(s^\dagger)\ge 2\delta$.
>
> Thus the corner solution is replaced by an interior delegation region, while the action regions remain governed by threshold level sets of $f_W(s^\dagger)$. Figure 1 of the PDF illustrates this interior region and the resulting partition of the $(\alpha,\beta)$ space.
>
> The current model treats $d$ instrumentally through outcomes and costs; incorporating intrinsic preferences over delegation levels is a natural extension outside the present specification.
>
> This supports the robustness of the regime structure. We will include this extension in the final version.
>
>
> >*“How would the analysis change if institutions could observe workflow choices and directly reward verification rather than only final outcomes?”*
>
> Thank you for the insightful question. To analyze the impact of directly rewarding verification, we extend the model by introducing a linear reward $R(s)=\gamma s$ provided by the institution to the worker. This changes the worker’s effective verification cost from $C_v(s)$ to $C_v(s)-\gamma s$, and modifies the verification surplus to
> $$
> \Phi_\gamma(s;\alpha,\beta)=\Phi(s;\alpha,\beta)+\gamma s.
> $$
> Let
> $$
> s^\dagger_\gamma(\alpha,\beta)\in\arg\max_{s\in[0,1]}\Phi_\gamma(s;\alpha,\beta),  \quad s^\dagger(\alpha,\beta)\in\arg\max_{s\in[0,1]}\Phi(s;\alpha,\beta).
> $$
> denote the optimal verification effort respectively. We can show that $s^\dagger_\gamma(\alpha,\beta)$ is increasing in $\gamma$, since additional reward $\gamma s$ encourages verification.
>
> An increasing $s^\dagger_\gamma(\alpha,\beta)$ represents that the worker is willing to put in greater verification and therefore changes the action-level regime boundaries (Figure 1(a) in the main paper). In particular, the boundary between pure delegation and verified delegation moves toward greater use of verification as $\gamma$ increases.
>
> Figure 3 in the attached PDF illustrates the resulting changes in the $(\alpha,\beta)$ partition, including the expansion of the verified-delegation regime and the corresponding shifts in the compliance-gain and quality-improvement regions.
>
> We will include this extension and the corresponding proposition in the final version.
>
>
> >*“The empirical section is not really validation but it is closer to a toy calibration on one dataset and one clinician. The jump from stylized theory to claims about workplace redesign feels a bit overclaim/too aggressive.”*
>
> Thank you for raising this point. We agree that the empirical section serves as an illustrative calibration rather than a rigorous validation. The goal of Section 4 is to demonstrate how commonly available institutional data can be mapped to the model’s parameters and used to evaluate worker quality and interventions in practice, rather than to establish general empirical claims.
>
> The single-clinician example is intended as an illustration of how the mapping procedure works in practice, not as a representative or statistically validated result. We will clarify this limitation explicitly in the final version.
>
> We also agree that our original phrasing regarding workplace redesign was too strong, and will revise it to better reflect the scope of the analysis.

---

> > ### Author Rebuttal · Reviewer_AHT4 · 2026-04-04
> >
> > My questions have been resolved and will maintain my score at the moment.

---

### Decision · Program_Chairs · 2026-04-30

**Decision:**

Accept (regular)

**Comment:**

I am happy to recommend the paper for acceptance.

The review team is very positive about the paper and the strength of the contributions. Everyone agrees that the topic is timely and important. If there is one critique the review team agreed on, it is that the model and assumptions of the paper are relatively stylized. That said, the team also believes the stylized model to be very informative and to help isolate high-level insights, so I do not think this is a major concern, and may be a strength of the paper. However, I believe this paper should be accepted. Please make sure to take the reviewer feedback into account for a camera ready version.